# Application of the Viscoelastic Continuum Damage Theory to Study the Fatigue Performance of Asphalt Mixtures—A Literature Review

Andrise Klug *, Andressa Ng and Adalberto Faxina

Department of Transportation Engineering, Sao Carlos School of Engineering, University of Sao Paulo, Sao Carlos 13566-590, Brazil; andressang@alumni.usp.br (A.N.); adalberto@usp.br (A.F.)
* Correspondence: andrise@usp.br; Tel.: +55-53984261832

**Abstract:** A viscoelastic continuum damage (VECD) model, which accounts for the effects of rate-dependent damage growth, has been successfully applied to quantify the reduction in the material integrity as a function of damage accumulation (characteristic curve) of asphalt mixtures. This paper presents an overview of the fundamentals of the VECD model and its simplifications (S-VECD) applied to the damage characterization of asphalt mixtures. It also presents a laboratory study carried out to evaluate the effects of the addition of reclaimed asphalt pavements (RAP) and a new binder (PG 64-22 or PG 58-16) on the fatigue performance of fine aggregate matrices (FAMs), in which the S-VECD theory was applied to analyze the results. The addition of RAP increased the stiffness and reduced the relaxation rate, resulting in FAMs that were stiffer and more susceptible to damage at high strain levels. The FAMs' fatigue factors ($FF_{FAM}$) indicated that the increase in RAP from 20% to 40% decreased the fatigue life of the mixtures. A strict control of the mixture variables is required, since the intrinsic heterogeneity of asphalt mixtures can lead to different characteristic curves for the same material.

**Keywords:** viscoelastic continuum damage theory; asphalt mixture; fine aggregate matrix; damage; reclaimed asphalt pavements

## 1. Introduction

An accurate prediction of the fatigue behavior of an asphalt mixture has been the goal of many studies focused on improving the asphalt mixture's design and the flexible pavement's performance [1–9]. Phenomenological models, which relate the stress or strain in the specimen with the number of cycles to failure, are simple tools to determine the fatigue behavior of an asphalt mixture in the laboratory [1,10–13]. However, this approach does not account for the complexity of the fatigue phenomenon [3]. More recently, mechanistic approaches have been employed rather than the phenomenological ones, as mechanistic models account for how damage evolves throughout the fatigue life at different loading and environmental conditions, leading to a better estimation of the fatigue behavior of the asphalt mixture [2,3,5,7].

Studies on the response of asphalt mixtures to fatigue cracking are divided into two categories: (1) the full asphalt mixture, which comprised asphalt binder, coarse aggregate, fine aggregate, and mineral filler; and (2) the fine aggregate matrix (FAM), composed of fine aggregate, mineral filler, and asphalt binder [14]. Based on the premise that the changes in the material microstructure are the beginning of the fatigue process [4,14,15], many researchers have studied the fatigue behavior of asphalt mixtures using the FAM approach [5,6,14,16–18]. The fine aggregate matrix represents an intermediate scale between the asphalt mastic and the full asphalt mixture, presenting an internal structure that is not affected by the coarse aggregate particles [19]. Studies at the FAM scale might be capable of providing a more realistic characterization of the fatigue response of full asphalt

mixtures than the one provided by tests performed on the mastic [14]. Another advantage of using FAMs in fatigue testing is that the reduced size of the specimens requires a smaller amount of material, as compared with the amount needed to produce specimens of full asphalt mixtures, which also reduces the laboratory work needed in the preparation of specimens [20].

Asphalt concrete is a material with high viscoelasticity imparted by the binder matrix [21]. The data obtained in tests performed with both FAMs and full asphalt mixtures can be analyzed by means of the continuum mechanics theory. According to the viscoelastic continuum damage (VECD) theory, the time dependency of viscoelastic materials can be eliminated by means of correspondence principles, which transform physical variables (stress, strain, and stiffness) in pseudo variables (pseudo stress, pseudo strain, and pseudo stiffness) [15,22,23]. According to this theory, a damaged body presenting internal microcracks is assumed to be an undamaged body with a reduced pseudo stiffness (C), with the microcracks uniformly distributed within the body [22,23]. The material damage evolution is described as the function C(S), in which a reduction in pseudo stiffness (C) is related to a material internal state variable (S) [15,21–23].

This review paper aims to provide an overview of the fundamentals of the viscoelastic continuum damage theory applied to the damage characterization of asphalt mixtures at the scales of both full asphalt mixture and fine aggregate matrix and its simplifications, along with an overview of the mechanistic fatigue life prediction model based on continuum mechanics. In addition to this review, this paper also presents examples of laboratory tests performed to evaluate the effect of reclaimed asphalt pavements and recycling agents on the fatigue response of FAMs. Although the theory presented in this paper assumes that each material presents a single characteristic curve that is independent of the loading conditions, laboratory tests sometimes result in different characteristic curves for specimens of the same FAM. Material heterogeneity is supposed to be the reason for distinct characteristic curves for specimens of the same FAM. In an attempt to reduce the material variability, the method adopted in this study employed the average of the linear-viscoelastic properties of at least three specimens to build a unique characteristic curve of the material, following the procedure employed to perform damage tests on full asphalt mixture specimens. Figure 1 provides an overview of the topics covered in this paper.

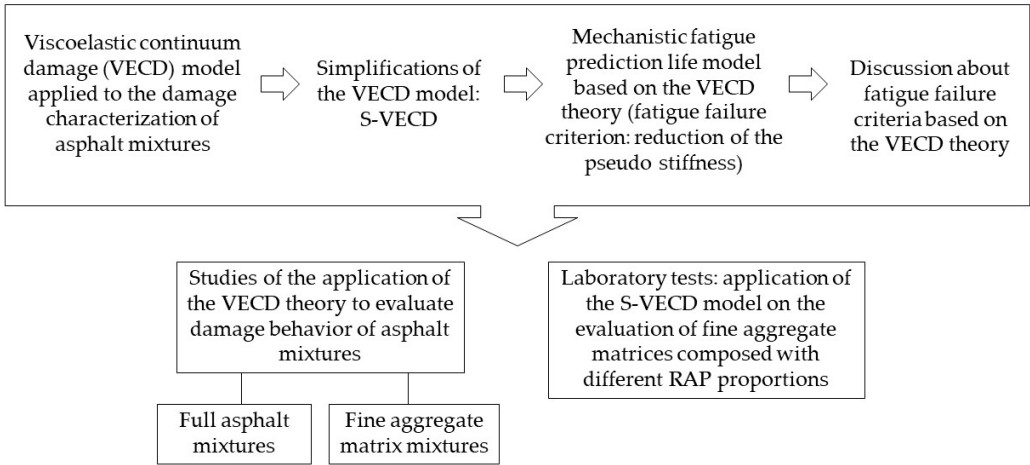

**Figure 1.** Overview of the paper's main topics.

## 2. Theory of Viscoelastic Continuum Damage

The work potential theory, which is based on the methods of thermodynamics of irreversible process, was developed by Schapery [15–23] in order to describe the mechanical behavior of elastic materials with growing damage. The theory characterizes the material using macroscale observations, quantifying the changes of the material microstructure by means of internal state variables (S). The elastic model was extended to describe the

mechanical behavior of viscoelastic media, by means of elastic–viscoelastic correspondence principles that eliminate the time dependence of the viscoelastic material.

Park et al. [21], Lee [24], and Lee and Kim [25] applied Schapery's theory to develop a constitutive model that describes the damage evolution process of asphalt mixtures for different materials, as well as loading and environmental conditions. This constitutive model was simplified by Lee et al. [7] to build a practical fatigue prediction model for specimens of asphalt mixtures under uniaxial cyclic loading. Such a model was later adapted by Kim and Little [5] for specimens of sand asphalt mixtures under torsional shear without rest periods. This improved model is considered capable of providing a reasonable representation of the fatigue life of asphalt mastics and fine aggregate matrices [5].

*2.1. The Work Potential Theory*

According to the thermodynamic theory, a generalized notation of the behavior of an elastic body that changes its structure is expressed by means of relationships between generalized forces, $Q_j$, and independent generalized displacements, $q_j$, as shown in Equation (1), where $\delta q_j$ is the virtual displacement and $\delta W'$ is the virtual work. For different physical situations, $q_j$ can represent strain, displacement, or rotation; and $Q_j$ can be stress, force, or moment. For any process of interest, the existence of a strain energy function, $W = W(q_j, S_m)$, was assumed, where $S_m$ (m = 1, 2, 3, M) refers to the increase in the value of the internal state variable, S. The relationship between the work done on a body during the process in which damage occurs and the strain energy function is expressed by Equation (2), where $f_m$ is the thermodynamic force (Equation (3)).

$$\delta W' = Q_j \delta q_j \tag{1}$$

$$dW = \frac{\partial W}{\partial q_j} dq_j + \frac{\partial W}{\partial S_m} dS_m = Q_j dq_j - f_m dS_m \tag{2}$$

$$f_m = -\frac{\partial W}{\partial S_m} \tag{3}$$

By integrating Equation (2) during a time interval $t_1$–$t_2$, and assuming that a state function $W_S = W_S(S_m)$ exists such that the thermodynamic force, $f_m$, is given by Equation (4), the work to vary the internal state of the material from a state 1 to a state 2 is expressed by Equation (5). The variable $\dot{S}_m$ is the damage evolution rate. By solving the integral in Equation (5), the work is given by Equation (6). Assuming that the time $t_1 = 0$, the total work from t = 0 to the current time $t_2$ is given by Equation (7).

$$f_m = \frac{\partial W_S}{\partial S_m} \text{ when } \dot{S}_m \neq 0 \tag{4}$$

$$\Delta W_T = W^{(2)} - W^{(1)} + \int_1^2 f_m dS_m \tag{5}$$

$$\Delta W_T = W^{(2)} - W^{(1)} + W_S^{(2)} - W_S^{(1)} \tag{6}$$

$$W_T = W + W_S \tag{7}$$

The total work input to the elastic body by the forces $Q_j$ is given by Equation (8), where j = 1, 2, . . . , J, and $W_T$ is the total work done on the body while considering that $S_m$ is variable in time. The elements of Schapery's theory, expressed in terms of stress–strain relationships, can be represented as follows in Equations (9) and (10), where $\sigma_{ij}$ is the stress tensor, $\varepsilon_{ij}$ is the strain tensor, and $S_m$ is the internal state variable. By considering Equations (3) and (4), the damage evolution law for the elastic media is represented by Equation (11), in which $W_S = W_S(S_m)$ is the dissipated energy due to damage growth. The right-hand side of the law represents the required force for damage growth, while the

left-hand side of the damage evolution law represents the available thermodynamic force to produce damage growth.

$$W_T = \int Q_j dq_j \tag{8}$$

$$W = W(\varepsilon_{ij}, S_m) \tag{9}$$

$$W_T = \int \sigma_{ij} d\varepsilon_{ij} \tag{10}$$

$$-\frac{\partial W}{\partial S_m} = \frac{\partial W_S}{\partial S_m} \tag{11}$$

*2.2. Elastic–Viscoelastic Correspondence Principle*

As demonstrated by Schapery [22], by using correspondence principles and a new damage evolution law, the elastic continuum damage theory can be extended to describe damage evolution in viscoelastic materials. For elastic materials, the stress–strain relationship is expressed by Hooke's law (Equation (12)), where $\sigma$ is the stress, E is the elasticity modulus, and $\varepsilon$ is the strain. For viscoelastic materials, the time dependency has to be considered, and the stress is expressed by a convolution integral (Equation (13)), where $\tau$ is an increment in the value of time t, G(t) is the material relaxation modulus, and $\varepsilon$ is the time-dependent strain in a viscoelastic material.

$$\sigma = E\varepsilon \tag{12}$$

$$\sigma = \int_0^t G(t - \tau)\frac{d\varepsilon}{d\tau}d\tau \tag{13}$$

The artifice proposed to eliminate the time-dependent effects was to transform the stress–strain relationships of the viscoelastic media into a pseudo domain that corresponds to a hypothetical elastic material, suggesting that the constitutive equation for viscoelastic media (Equation (14)) is identical to that for elastic media (Equation (12)). However, the stress and the strain are not necessarily physical quantities, but pseudo variables: pseudo stress ($\sigma^R$) and pseudo strain ($\varepsilon^R$). According to the second correspondence principle (CP II), $\sigma^R = \sigma$, where $\sigma$ is the time-dependent stress applied to a viscoelastic material; and the pseudo strain is given by Equation (15), where $\varepsilon$ is the time-dependent strain in a viscoelastic material, G(t) is the linear-viscoelastic relaxation modulus of the material, and $E^R$ is the modulus of hypothetical elastic material [22,23].

$$\sigma = E_R \varepsilon^R \tag{14}$$

$$\varepsilon^R = \frac{1}{E_R} \int_0^t G(t - \tau)\frac{d\varepsilon}{d\tau}d\tau \tag{15}$$

The same equations for elastic materials are used to solve viscoelastic cases by considering elastic–viscoelastic correspondence principles. Equation (9) is then expressed by Equation (16), the pseudo strain energy density ($W^R$) function, where the physical strain, $\varepsilon$, is substituted by the pseudo strain, $\varepsilon^R$, and $S_m$ is the internal state variable. The stress-pseudo strain relationship is given by Equation (17), where $\sigma$ is the stress, $W^R$ is the pseudo strain energy density, and $\varepsilon^R$ is the pseudo strain [23]. For most viscoelastic media, the available force for damage growth and the resistance against the growth are rate-dependent. For this reason, the damage evolution law for elastic materials (Equation (11)) cannot just be transformed into a damage evolution law for viscoelastic materials by the use of correspondence principles without further modification. The new damage evolution law for viscoelastic materials is given by Equation (18), where $S_m$ is the damage evolution

rate, $W^R$ is the pseudo strain energy density, $S_m$ is the internal state variable, and $\alpha_m$ is a material-dependent constant [26].

$$W^R = W^R\left(\varepsilon^R, S_m\right) \tag{16}$$

$$\sigma = \frac{\partial W^R}{\partial \varepsilon^R} \tag{17}$$

$$\dot{S}_m = \left(-\frac{\partial W^R}{\partial S_m}\right)^{\alpha_m} \tag{18}$$

*2.3. Viscoelastic Continuum Damage Model Applied to Asphalt Mixtures*

Kim and Little [27] successfully applied Schapery's [15,22] theory on the development of a nonlinear-viscoelastic constitutive equation that should represent the damage growth in asphalt concrete. The equation was developed from tests on mixtures of asphalt concrete composed of crushed granite fines and asphalt binder, and it is presented by Equation (19), where $\sigma$ = stresses in a body; $\varepsilon^R$ = strains in a body (Equation (15)); $\varepsilon_L^R$ = maximum pseudo strain in the past history; and $S_p$ = damage parameter [28]. The form of the damage parameter based on pseudo strain is shown as follows in Equation (20), where $p = (1 + N)k$; $N$ = the exponent of the power law between stress and strain, $\sigma \sim |\varepsilon^R|^N$; $k = 2(1 + 1/m)$; $m$ = the exponent of the power law between creep compliance and time; and $D(t) = D_1 t^m$. When repetitive loading is applied, numerical integration can be used to obtain $S_p$, as shown in Equation (21), assuming that $d\varepsilon^R/dt$ is constant within the range of the experimental data points. A uniaxial tensile testing was employed to generate all data for the construction of the constitutive equation, which satisfactorily predicted effects due to multilevel loading, the sequence of multilevel loading, and various durations of rest periods. From the tests, it was proved that the history dependence of asphalt concrete with negligible damage growth can successfully be eliminated by the CP II, $\sigma^R = \sigma$.

$$\sigma = \sigma\left(\varepsilon^R, \varepsilon_L^R, S_p\right) \tag{19}$$

$$S_p = \left(\int_0^t |\varepsilon^R|^P dt\right)^{\frac{1}{P}} \tag{20}$$

$$[S_p(t)]^P = \int_0^{t_{j-1}} |\varepsilon^R|^P dt - \int_{t_{j-1}}^t |\varepsilon^R|^P dt \tag{21}$$

Park et al. [21] also proposed a uniaxial viscoelastic damage model to characterize the behavior of asphalt mixtures with time-dependent damage growth, subjected to different strain rates under uniaxial stress. The pseudo strain energy density ($W^R$) function is given by Equation (22), where $C$ is a function of the damage parameter $S$, and $\varepsilon^R$ is the pseudo strain. For linear-viscoelastic behavior and fixed damage, the stress, $\sigma$, can be written as in Equation (23). The damage evolution law (Equation (18)) is reduced to the single equation for $S$ (Equation (24)), where $S$ is the damage parameter, $\alpha$ is a material-dependent constant, and the over dot denotes a time derivative.

$$W^R = \frac{1}{2}C(S)\left(\varepsilon^R\right)^2 \tag{22}$$

$$\sigma \equiv \frac{\partial W^R}{\partial \varepsilon^R} = C(S)\varepsilon^R \tag{23}$$

$$\dot{S}_m = \left(-\frac{\partial W^R}{\partial S}\right)^{\alpha} \tag{24}$$

By substituting Equation (22) into Equation (24), Park et al. [21] obtained a relationship between S and a new damage parameter, S∗, given by Equation (25), where S∗ is a function of the strain history (Equation (26)). The variable k is a free constant considered in their study, such that the maximum values of S and S∗ are numerically equal. Thus, Equation (23) was replaced by Equation (27). In Equations (25)–(27), $\alpha$ is a material-dependent constant, $\varepsilon^R$ is the pseudo strain, and C is a function of the damage parameter S.

$$S^* = k\left[\int_0^S \frac{dS}{(-0.5dC/dS)^\alpha}\right]^{1/(2\alpha)} \tag{25}$$

$$S^* \equiv k\left[\int_0^t |\varepsilon^R|^{2\alpha} dt\right]^{1/(2\alpha)} \tag{26}$$

$$\sigma = C(S^*)\varepsilon^R \tag{27}$$

The function C(S) and the $\alpha$ value must be determined. However, by fitting Equation (23) to an experimental stress–pseudo strain curve, it is possible to obtain a C modulus that depends on $\varepsilon^R$ and strain rate, but not on S. In order to find the dependence of the C modulus on S, the use of the damage evolution law (Equation (18)) is required. However, the evolution law itself requires prior knowledge of C(S), making this procedure inefficient to find C and its dependence on S. The method proposed by Park et al. [21] to overcome this problem was to determine a transformed damage variable, Ŝ (Equation (28)), that may be obtained from the numerical scheme presented in Equation (29), where $\varepsilon_i^R$(i = 1, 2, 3, N) denotes pseudo strain levels, C′ ≡ dC/dŜ, and Ŝ(0) = 0. This method allows one to obtain the function C(Ŝ) from experimental stress–pseudo strain curves, and then the function C(S) can be obtained from Equation (28), by replacing the transformed damage variable Ŝ for the original damage variable, S, where $\alpha$ is a material-dependent constant.

$$\hat{S} \equiv \frac{1}{(1+1/\alpha)} S^{(1+1/\alpha)} \tag{28}$$

$$\hat{S}\left(\varepsilon_{i+1}^R\right) = \hat{S}\left(\varepsilon_i^R\right) + \frac{C\left(\varepsilon_{i+1}^R\right) - C\left(\varepsilon_i^R\right)}{\Delta\hat{S}} \tag{29}$$

The constant $\alpha$, related to the material's creep or relaxation properties, also must be determined. According to Schapery [15], depending on the characteristics of the failure zone on a crack tip, $\alpha$ is expressed by Equation (30), if the material's fracture energy and failure stress are constant, or $\alpha$ is expressed by Equation (31), if the fracture process zone size and the material's fracture energy are constant, where the material relaxation rate, m, is given by Equation (32), where D(t) = uniaxial creep compliance, E(t) = uniaxial relaxation modulus, and t = time. Park et al. [21] employed the relationship presented in Equation (30) by assuming that the damage in the specimen is closely related to the growth of microcracks. The constant $\alpha$ was obtained by a method of successive approximations until the model better fit the experimental observations.

$$\alpha = \left(1 + \frac{1}{m}\right) \tag{30}$$

$$\alpha = \frac{1}{m} \tag{31}$$

$$m = \frac{\log D(t)}{\log(t)} \text{ or } m = \frac{-\log E(t)}{\log(t)} \tag{32}$$

Lee and Kim [24,25] also proposed a solution to the damage evolution law by studying the mechanical behavior of asphalt concrete. They conducted uniaxial tensile cyclic loading tests with different loading amplitudes, under controlled-strain and controlled-stress modes of loading. The researchers observed a decrease in the slope of each σ–$\varepsilon^R$

cycle as the number of loading increased, and they found it necessary to define the secant pseudo stiffness, $S^R$, to represent this change in the slope of stress–pseudo-strain loops (Equation (33)), where $\varepsilon_m^R$ is the peak pseudo strain in each stress–pseudo-strain cycle and $\sigma_m$ is the stress that corresponds to $\varepsilon_m^R$. To minimize the sample-to-sample variability, the pseudo stiffness was divided by the initial pseudo stiffness, I, resulting in the normalized pseudo stiffness, C, represented by Equation (34). By considering Equation (33) and Equation (34), the constitutive equation for viscoelastic materials with growing damage is expressed by Equation (35). The normalized pseudo stiffness, $C(S_m)$, is a function of the internal state variables, $S_m$, and represents the microstructural changes of the body. The researchers assumed an internal state variable, $S_1$, to determine the change in pseudo stiffness due to growing damage, and the work function ($W^R$) for viscoelastic materials is given by Equation (36), where $C_1(S_1)$ is a function that represents $S^R$.

$$S^R = \frac{\sigma_m}{\varepsilon_m^R} \tag{33}$$

$$C = \frac{S^R}{I} \tag{34}$$

$$\sigma = IC(S_m)\varepsilon_m^R \tag{35}$$

$$W^R = \frac{I}{2}C_1(S_1)\left(\varepsilon_m^R\right)^2 \tag{36}$$

Despite that the material function $C_1$ can be determined by using experimental data and the damage evolution law (Equation (18)), this procedure is not convenient to find $C_1$ and its dependence on $S_1$, because the evolution law requires prior knowledge of $C_1(S_1)$. The method presented to overcome this problem was to use a chain rule (Equation (37)) to eliminate the S on the right-hand side of the evolution law, and by means of mathematical substitutions (Equation (38)) the numerical approximation given by Equation (39) was obtained, where $\varepsilon^R$ is the pseudo strain, t is the time, I is a factor to normalize the pseudo stiffness, and $\alpha$ is a material-dependent constant (Equations (30) and (31)). The function $C_1(S_1)$ can be obtained by cross-plotting the C values obtained from Equation (35) against the S values obtained from Equation (39), and by performing a regression on the data (Equation (40)), where $C_{10}$, $C_{11}$, and $C_{12}$ are regression coefficients; or an exponential function (Equation (41)), where a and b are the calibration constants.

$$\frac{dC}{dS} = \frac{dC}{dt}\frac{dt}{dS} \tag{37}$$

$$\frac{dS}{dt} = \left[-\frac{1}{2}\frac{dC}{dS}\left(\varepsilon^R\right)^2\right]^{\frac{\alpha}{(1+\alpha)}} \tag{38}$$

$$S \equiv \sum_{i=1}^{N}\left[\frac{I}{2}\left(\varepsilon^R\right)^2(C_{i-1}-C_i)\right]^{\alpha/(1+\alpha)}(t_i-t_{i-1})^{1/(1+\alpha)} \tag{39}$$

$$C_1(S_1) = C_{10} - C_{11}(S_1)^{C_{12}} \tag{40}$$

$$C_1(S_1) = \exp\left(aS_1^{\,b}\right) \tag{41}$$

For the material parameter $\alpha$, Lee and Kim [29] observed that Equation (30) was adequate for the controlled strain mode of loading, while Equation (31) was a better assumption for the controlled stress mode of loading. These observations suggested that the material's fracture energy and failure stress are constant under the controlled strain mode, whereas the material's fracture energy and the fracture process zone size are constant under the controlled stress mode.

### 2.4. Pseudo-Strain Calculation

The constant cyclic shear strain, $\varepsilon(t)$, as a function of time, can be simply represented as an analytical harmonic sinusoidal function (Equation (42)), where $\varepsilon_0$ is the shear strain amplitude, $\omega$ is the angular velocity, $\theta$ is a regression constant, and H(t) is the Heaviside step function. Substituting Equation (42) within the definition of pseudo strain (Equation (15)) and assuming that $E_R$ is 1, the pseudo strain at the current time can be analytically represented by Equation (43), where $\phi$ is the phase angle and $|G^*|$ is the linear-viscoelastic dynamic shear modulus. As can be seen, the pseudo strain at any time can be predicted with a well-defined strain history as a function of time and two material properties: dynamic modulus and phase angle. In the study of fatigue, only the peak pseudo strain within each cycle is used, and the pseudo strain reaches the peak pseudo strain in each cycle when the sine function in Equation (43)) becomes 1, as represented by Equation (44)), where $\varepsilon_m^R$ is the peak pseudo strain at each cycle [8,30].

$$\varepsilon(t) = \varepsilon 0 \, \sin \, (\omega t + \theta) \, H(t) \tag{42}$$

$$\varepsilon^R(t) = \varepsilon 0 \, |G^*| \, \sin \, (\omega t + \theta + \phi) \tag{43}$$

$$\varepsilon_m^R(t) = \varepsilon 0 \, |G^*| \tag{44}$$

### 2.5. Simplified Viscoelastic Continuum Damage Model

Different researchers have worked on developing a simplified viscoelastic continuum damage (S-VECD) model in order to characterize asphalt mixtures in a quick and easy way using cyclic fatigue test data [9,30–35]. The modification in the continuum damage equations [8,15,21–25,27,29] proposed by Christensen and Bonaquist [31] can be used to predict the fatigue behavior of asphalt concrete mixtures based on their volumetric composition and degree of compaction, and the rheological type of the asphalt binder used in the mixture. In a later study, Christensen and Bonaquist [32] presented two new concepts for inclusion in the continuum damage analysis of fatigue data on hot-mix asphalt (HMA) mixtures: the concept of reduced loading cycles and the concept of effective strain. Kutay et al. [30] developed and validated a simple equation for calculation of the continuum damage parameter (S) for the specific case of fixed-frequency cyclic tests. Underwood et al. [33,34] reviewed and discussed the limitations of the simplified models proposed by Kutay et al. [30], and Christensen and Bonaquist [31] suggested an improved S-VECD model.

The original application of the VECD model to cyclic data requires that the pseudo strain, pseudo stiffness, and damage are calculated and tracked for the entire loading history. Although the analysis of tests with a large number of data points is not impossible with modern computers, it depends on advanced computational schemes. Further, experimental difficulties related to data storage and electrical interference (noise and phase distortion) can lead to significant errors. In an effort to minimize these shortcomings, simplified mechanics analyses were developed by researchers at North Carolina State University (NCSU) and by the Federal Highway Administration (FHWA). In the methods developed at NCSU, the pseudo strain is calculated for the entire loading history (Equation (15)), and present the same sequence of calculations for damage during the first load path (Equation (39)). This early portion of the damage calculation is referred to as the transient calculation. After the first loading cycle, however, the sequence of calculations is referred to as cyclic calculations, and a simplified calculation method is used. The NCSU simplified methods also consider the specimen-to-specimen variation by the I factor, which is defined by the initial slope of the stress–pseudo-strain plot obtained in a stress-controlled test within the linear–viscoelastic (LVE) range. In some of the NCSU simplified models, the permanent pseudo strain ($\varepsilon_S^R$) is also taken into account for the damage calculation [33,34]. The main difference between the NCSU S-VECD models and the one proposed by the FHWA is the simplification of the pseudo strain calculation (Equation (45)). This simplification is used as reference for the steady-state assumption, because it is considered theoretically accurate

for the zero-mean-stress steady-state conditions. In addition, in the FHWA S-VECD model, the pseudo strain is calculated based on the peak-to-peak values, and assuming that the damage growth occurs during the tension and compression load [33,34].

$$\varepsilon_0^R = \varepsilon_0 |G^*|_{LVE} \tag{45}$$

Characteristic curves of five asphalt mixtures built by using the NCSU S-VECD models and the FHWA S-VECD model were compared. Based on the results of this comparison, a new simplified model was developed while considering the advantages of each S-VECD model [33,34]. Taking into account the modified correspondence principle [22] and zero-mean-stress steady-state condition considered in the FHWA S-VECD, a new S-VECD model was proposed [33,34]. The relationships for pseudo-strain amplitude, pseudo-stress amplitude, and pseudo stiffness at the $k^{th}$ load cycle for a stress-controlled cyclic test were simplified, as shown in Equations (46) and (47). In this FHWA S-VECD method, damage is expected to grow according to Equation (48).

$$\sigma_{0,k}^P \sigma_{0,k} \tag{46}$$

$$C_k(S) \equiv \frac{\sigma_{0,k,pp}^R}{I \cdot \varepsilon_{0,k,pp}^R} = \frac{|G^*|_k}{I \cdot |G^*|_{LVE}} \tag{47}$$

$$dS \equiv \left[ \frac{I}{2} \left( \varepsilon_{0,k,pp}^R \right)^2 (C_{k-1} - C_k) \right]^{\alpha/(1+\alpha)} (t_k - t_{k-1})^{1/(1+\alpha)} \tag{48}$$

The parameters $\sigma_{0,k}$, $\sigma_{0,k}^R$, $\varepsilon_{0,k}^R$, $|G^*|_k$, and $|G^*|_{LVE}|$ represent, respectively, the shear stress amplitude, the shear pseudo-stress amplitude, the shear pseudo-strain amplitude, the dynamic shear modulus at the $k^{th}$ loading cycle, and the linear-viscoelastic dynamic shear modulus. The comma subscript, pp, denotes the pseudo strain computed based on peak-to-peak values. The I factor is used as a correction factor in order to normalize test results while considering the sample-to-sample variability of the initial dynamic shear modulus. Hou [9] conducted a study in order to verify the S-VECD model proposed by Underwood et al. [33] by applying it to various types of asphalt concrete mixtures under various conditions. Hou [9] observed that the model could be applied to accurately predict the fatigue life of asphalt concrete under cyclic loading at multiple temperatures and strain levels. Lee et al. [36] recommended that the power law model (Equation (40)) should be fitted to the characteristic curves obtained from the S-VECD model after filtering the data to produce damage (S) increments of 5000. Lee et al. [36] also proposed a shape factor, which is calculated as the ratio of the area above the damage characteristic curve for an individual test replicate (i.e., $A_{measured}$) to the area above the fitted model damage characteristic curve obtained by Equation (40) (i.e., $A_{predicted}$). Replicates with shape factors greater than 1.1 or lower than 0.9 are considered outliers and should not be considered for analysis.

### 2.6. Mechanistic Fatigue Life Prediction Model

There are two main approaches to characterize the material fatigue behavior in the laboratory: the phenomenological approach and the mechanistic approach. The phenomenological models relate the stress or strain in the specimen with the number of cycles to failure. This approach is relatively simple, but it does not account for the complexity of the fatigue phenomenon. On the other hand, the mechanistic approach accounts for how the damage evolves throughout the fatigue life at different loads and environmental conditions, leading to a better estimation of the fatigue behavior of asphalt mixtures.

The mechanistic VECD fatigue prediction model has been applied as a modern method to estimate the mechanical behavior of asphalt mixtures. Different versions of the formulations have been developed by various researchers [5,7,37,38]. In this paper, the model developed by Lee et al. [7] and Kim and Little [5] is presented, and it will be applied to compare the fatigue lives of the FAMs evaluated in this study. The model estimates the number

of cycles required to degrade the material, $N_f$, to a certain pseudo-stiffness level, C, or to reach a certain amount of damage, $S_f$, at an arbitrary frequency, f, and controlled pseudo-strain amplitude, $\varepsilon^R$. The parameters $C_{11}$ and $C_{12}$ are obtained from Equation (40) [5,7]. Kim and Little [5] performed torsional shear cyclic tests in sand asphalt mixtures, and compared the measured fatigue lives of the materials with the values predicted from the fatigue model (Equations (49)–(51)). They concluded that the model parameters might provide a reasonable representation of the fatigue response.

$$A = f \left\{ \frac{1}{2} C_{11} C_{12} \right\}^\alpha \{1 + \alpha(1 - C_{12})\}^{-1} S_f^{[1+\alpha(1-C_{12})]} \tag{49}$$

$$B = 2\alpha \tag{50}$$

$$N_f = A[\varepsilon_R]^{-B} \tag{51}$$

### 2.7. Fatigue Failure Criterion

Studies have shown that the traditional fatigue failure relationship based on constant amplitude loading [39], used in a variety of layered elastic pavement design methods, may underestimate field fatigue life by as much as 100 times [40]. This difference between laboratory and field fatigue performance may be attributed to differences between laboratory and field loading conditions, such as rest periods and varying load magnitude in the field. By applying the VECD approach, it was observed that fatigue failure was influenced by both relaxation and damage mechanisms, and due to this, the traditional failure criterion (50% reduction in the initial stiffness) was modified to the 50% reduction in the initial secant pseudo-stiffness criterion [25]. However, a 50% reduction in the initial secant pseudo-stiffness may not represent the fatigue failure of all materials [5,41,42]. Other criteria based on changes in phase angle during fatigue testing and the reduced energy ratio concept have been defended in studies by Rowe [10], Reese [11], Hopman et al. [12], and Rowe and Bouldin [13].

Research efforts have been conducted to identify a proper failure criterion based on the VECD approach and energy-based concept [5,43–49]. Daniel et al. [43] made a comparison between the VECD and dissipated energy (DE) approaches using uniaxial direct-tension fatigue tests. The two approaches were also compared to the traditional phenomenological approach relating the initial strain to the number of cycles to 50% reduction in initial stiffness. The number of cycles to failure showed strong correlation between the VECD and the DE failure criteria. Findings of a study conducted by Kim and Little [5] showed that three damage parameters were effective to characterize fatigue damage during torsional loading: (i) a decay in pseudo stiffness; (ii) a loss of nonlinear dynamic modulus; and (iii) a change in dissipated strain energy.

In a study by Hou [9], the pseudo stiffness at failure was plotted against test reduced frequency for multiple mixtures to provide an empirical observation of all the tested mixtures, in order to determine the failure criterion. It was found that: (i) the pseudo stiffness at failure increased with the reduced frequency, and (ii) the rate of change in the pseudo stiffness at failure as a function of reduced frequency increased. The observations led to a reduced-frequency piecewise function that was applied for failure criterion development. Tarefder et al. [44] conducted a study in order to compare the traditional stiffness [50] and the energy-based fatigue failure criterion [13] to the fatigue failure criterion based on the VECD approach [40]. It was shown that the traditional approach (50% reduction in the initial stiffness) was conservative, and that the fatigue life predicted by the VECD approach was always shorter than the one predicted by the energy ratio approach.

Sabouri and coworkers proposed a new energy-based failure criterion for the continuum damage model that should consistently predict failure of a material that reaches macrofracture. The proposed failure criterion relates the average of the dissipated pseudo energy values in cycles, denoted as $G^R$, and the number of cycles to fatigue failure. The

criterion was able to predict the fatigue life of asphalt concrete mixtures across different temperatures and strain amplitudes [45,46], and has been applied to predict fatigue behavior of reclaimed asphalt pavement (RAP) mixtures and non-RAP mixtures, mixtures prepared with modified and unmodified binders, warm-mix asphalt mixtures, and long-term aged mixtures [51–60]. Keshavarzi and Kim [49] extended the $G^R$ criterion concepts to determine when fracture occurs in monotonic failure tests, such as thermal stress restrained specimen tests (TSRSTs).

Mensching et al. [61] suggested that more studies are needed to determine a $G^R$ index that is more representative of field conditions, because there was not a clear trend between RAP increase and a change in the mixture fatigue performance when the damage characteristic curves and $G^R$ failure criterion were used. More recently, Wang and Kim [48] developed and validated a new energy-based failure criterion based on the S-VECD model ($D^R$). The advantages of the $D^R$ failure criterion in comparison to the previous $G^R$ failure criterion are: (i) the $D^R$ can be computed for each fatigue test and then used to check the sample-to-sample variability for each test, (ii) $D^R$ is obtained in arithmetic scale rather than in log–log scale, and (iii) the number of tests needed to obtain $D^R$ is fewer than for the $G^R$ failure criterion [48]. In a study by Wang et al. [62], a three-dimensional finite element program (FlexPAVETM) was used to simulate the fatigue performance of field test sections. The fatigue damage of the sections was predicted using the FlexPAVETM software by considering both $G^R$ and $D^R$ criteria. The $D^R$ failure criterion was found to yield more realistic fatigue-cracking performance predictions than the $G^R$ failure criterion [62], and has been checked for RAP and aged mixtures [63,64].

Etheridge et al. [47] studied the correlations among $D^R$, an index parameter referred to as apparent damage capacity ($S_{app}$) [65], and mix design factors such as nominal maximum aggregate size (NMAS), asphalt binder type, and binder content. The $S_{app}$ parameter was shown to be able to predict the fatigue-cracking propensy of asphalt mixtures [47,65,66], and it was found to have a strong relationship with polymer modification and the NMAS of asphalt mixtures [47]. Based on experimental data and the Georgia Department of Transportation's practical guidelines for specific mixtures, a study by Etheridge et al. [47] developed $S_{app}$ threshold values for different traffic levels. In a study by Zhang et al. [67], the $D^R$ parameter showed good correlation with three new performance indices from the linear amplitude sweep (LAS) test: (i) strain tolerance ($\varepsilon_T$), (ii) strain energy tolerance ($\varepsilon_E$), and (iii) average reduction in integrity to failure ($I^R$).

*2.8. Linear Viscoelasticity*

The linear-viscoelastic characterization of asphalt mixtures is carried out by measuring relaxation or creep properties. The creep and relaxation tests measure the response of the materials to a constant load or displacement over time; i.e., they describe the material properties in the time domain within the linear-viscoelastic region. In the creep test, a constant load is applied to the specimen over time at a constant temperature, and the creep compliance, D(t), is defined as the ratio of accumulated strain to the constant stress magnitude at a specific time. In the relaxation test, the specimen is submitted to a constant strain for a given period of time at a constant temperature, and the relaxation modulus, E(t), is defined as the ratio of the stress evolution to the constant strain magnitude. However, in some cases, it is not possible to obtain an accurate response of the material by carrying out a short-time test with transient excitation; e.g., static loading. To overcome this limitation, tests with steady-state sinusoidal excitation are adopted; e.g., dynamic loading.

Viscoelastic materials under dynamic loading conditions provide frequency-domain dynamic properties, such as (i) phase angle, $\phi(\omega)$, which represents the gap between the stress and strain due to the time-dependency of viscoelastic materials; (ii) storage shear modulus, $G'(\omega)$, which represents the elastic characteristics of the material; and (iii) loss shear modulus, $G''(\omega)$, which corresponds to the viscous behavior of the material. The combined form of storage shear modulus and loss shear modulus results in Equation (52) for the phase angle, where $\omega$ is the angular frequency; in Equation (53) for the dynamic

shear modulus, $|G^*(\omega)|$, where $\tau_{max}$ is the maximum shear stress at each cycle and $\gamma_{max}$ is the applied cyclic shear strain amplitude; and in Equation (54) for the complex shear modulus, $G^*(\omega)$, where i is equal to $\sqrt{(-1)}$.

$$\varphi(\omega) = \tan^{-1}\left[\frac{G''(\omega)}{G'(\omega)}\right] \tag{52}$$

$$|G^*(\omega)| = \frac{\tau_{max}}{\gamma_{max} = \sqrt{\left[G'(\omega)\right]^2 + \left[G''(\omega)\right]^2}} \tag{53}$$

$$G^*(\omega) = G'(\omega) + iG''(\omega) \tag{54}$$

A dynamic frequency sweep test within the linear-viscoelastic range is carried out in the dynamic shear rheometer (DSR) to define the linear-viscoelastic relaxation behavior of the material. A curve-fitting function for the linear-viscoelastic modulus and frequency is required to determine the linear-viscoelastic relaxation modulus from a test in the frequency domain.

The mechanical behavior of viscoelastic materials can be represented by an association of springs and dashpots. The spring represents the elastic behavior of the material, which obeys Hooke's law: $\sigma = G\varepsilon$, where $\sigma$ is the stress, $\varepsilon$ is the strain, and G is the elasticity modulus. The viscous behavior of the material is represented by the dashpot. Viscous elements are mathematically represented by Newton's law of viscosity: $\sigma = \eta\dot{\varepsilon}$, where $\eta$ is the viscosity coefficient and $\dot{\varepsilon}$ is the strain rate. Different arrangements of springs and dashpots can represent different material behaviors. The Maxwell model is a two-component model composed of a series connection of a spring and a dashpot, and corresponds to a viscoelastic fluid [68]. The Kelvin–Voigt Model is also a two-component model represented by a spring and a dashpot combined in a parallel fashion, and represents a viscoelastic solid [68,69]. Burger's model is a four-component model and consists of a series arrangement of a Maxwell and a Kelvin–Voigt element [69]. While the Maxwell and Burger's elements can exhibit typical rheological behavior of asphalt binders under constant stress–strain tests, the Kelvin element cannot describe relaxation and should not be used to represent asphalt binders in any hot-mix asphalt models [69].

In this study, the generalized Maxwell model, commonly referred to as the Prony series, is used as a curve-fitting function for the viscoelastic materials due to its capability of describing the different stages of the behavior of viscoelastic materials [70]. In the generalized Maxwell model, the Maxwell model elements are combined in a parallel fashion as shown in Figure 2. The relationship between the tensile stress $\sigma(t)$ and time is shown in Equation (55) [68].

$$\sigma(t) = \dot{\varepsilon}\eta(1 - e^{-\frac{Et}{\eta}}) \tag{55}$$

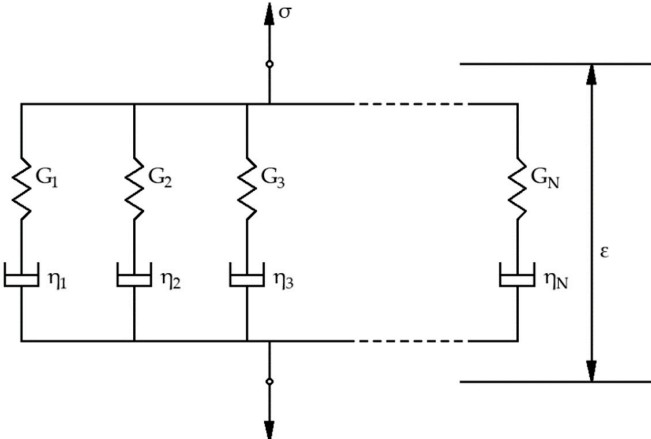

**Figure 2.** Generalized Maxwell model [68].

Prony series representation of storage and loss moduli as a function of frequency (frequency domain) was presented by Christensen [68], which is given by Equations (56) and (57), where $G_e$ is the equilibrium modulus, $G_i$ is the elastic modulus, $\rho_i$ is the relaxation time, $\omega$ is the angular frequency, and n is the number of elements of the Prony series needed to fit the analytical representation to the experimental data.

$$G'(\omega) = G_e + \sum_{i=1}^{n} \frac{G_i \omega^2 \rho_i^2}{\omega^2 \rho_i^2 + 1} \tag{56}$$

$$G''(\omega) = \sum_{i=1}^{n} \frac{G_i \omega \rho_i}{\omega^2 \rho_i^2 + 1} \tag{57}$$

The spring constants ($G_i$) are defined by means of the collocation method, a matching process between the analytical representation and the experimental data for a certain number of points. Considering the Prony series parameters found by the collocation method, the static relaxation shear modulus as a function of time (time domain) can be predicted from the dynamic shear modulus as a function of frequency (frequency domain) by Equation (58). The relaxation property (m-value) is determined as the slope of the relaxation modulus curve, in logarithm scale, and is used in the VECD approach to determine the damage evolution rate of the material. This material property can be obtained by adjusting a power law function (Equation (59)) to the relaxation curve predicted by the Prony series, where $G_0$ and $G_1$ are material constants, t is the time, and m is the slope of the relaxation curve in the time domain.

$$G(t) = G_\infty + \sum_{i=1}^{n} G_i e^{-\frac{t}{\rho_i}} \tag{58}$$

$$G(t) = G_0 + G_1 t^{-m} \tag{59}$$

The experimental characterization of the dynamic modulus of asphalt concrete mixtures follows the test protocol AASHTO TP 62 [71]. In the absence of a specific procedure for FAMs, researchers have been adopted different criteria to obtain the dynamic modulus within the linear-viscoelastic region of each material. For example, Ng [72] and Klug et al. [73] defined the linear-viscoelasticity region as the range of stresses under which the material presents a deviation of less than 10% of its initial stiffness, and a stress of 15 kPa was applied to the specimens to obtain the linear-viscoelastic properties. In turn, Karki et al. [16] subjected each specimen to an oscillatory strain amplitude of 0.008% in shear for 30 s at 10 Hz and 25 °C. Ideally, the linear-viscoelastic dynamic shear modulus, $|G^*|_{LVE}$, is equal to the dynamic shear modulus value that is measured by applying a small level of strain or stress within the linear-viscoelastic range of the material [16].

## 3. Studies on the Application of the VECD Theory

### 3.1. Full Asphalt Mixture Approach

Based on previous studies developed to characterize the damage behavior of asphalt mixtures [21,24,25,27,74], Daniel and Kim [75] proposed a testing procedure for fatigue characterization of asphalt concrete specimens under monotonic (constant-crosshead rate tests to failure) and cyclic (controlled-crosshead strain amplitude cyclic fatigue testing in tension) loading, which consists of two steps: (i) to perform a frequency sweep test on at least three replicate specimens at different frequencies and temperatures, for linear-viscoelastic material characterization (phase angle and relaxation modulus); and (ii) to perform a constant strain rate test to failure on all replicate specimens at a single rate at the desired temperature, for damage characterization. Pseudo strains are calculated according to Equation (15). Normalized pseudo stiffness, C, and damage parameter, S, for all times are calculated by Equation (34) and Equation (39), respectively. The values are cross-plotted to construct the characteristic curve that describes the reduction in material integrity

as damage grows in the specimen (CxS) and to determine the functional coefficients $C_1$ and $C_2$ (Equation (40)). Daniel and Kim [75] showed that a single CxS curve can be obtained for each material, regardless of the applied loading conditions (cyclic vs. monotonic, amplitude/rate, frequency). However, Lundström and Isacsson [76] indicated that it was difficult to generally predict fatigue results based on characteristic curves obtained from monotonic tests. A later study conducted by Keshavarzi and Kim [77] applied the viscoelastic continuum damage (VECD) theory to simulate asphalt concrete behavior under monotonic loading. In that study, direct tension monotonic testing that incorporated a constant crosshead displacement rate and four temperatures was used to simulate thermal cracking of asphalt concrete prepared with four reclaimed asphalt pavement (RAP) proportions. The predictions of monotonic simulation matched the measured data of the monotonic tests very well up to the point of maximum stress. More recently, Cheng et al. [66] observed that the asphalt mixture CxS curves were independent of the strain level, but affected by the loading waveform.

Daniel and Kim [75] also verified that the characteristic curve at any temperature below 20 °C can be found by utilizing the time–temperature superposition principle (t-TS) and the concept of reduced time. Later, Chehab et al. [78] demonstrated that t-TS can be extended from material's linear-viscoelastic range to high damage levels. Findings from compression tests [79,80] complemented the findings from Chehab et al. [79] for tension tests. Underwood et al. [81] also found that the t-TS principle with growing damage was applicable to mixtures with modified asphalt binders. The analysis of the fatigue behavior using the constant-crosshead rate test method and the t-TS principle [82] has been successfully employed to evaluate mixtures prepared with RAS, mixtures prepared with RAP, warm-mix asphalt (WMA) mixtures, mixtures prepared with modified binders, and other factors such as aggregate gradation, air voids, moisture, and aging [9,56–64,83–92].

In the studies carried out by Kim et al. [93] and Underwood et al. [33,35], a simplified VECD model was implemented in a finite element package (FEP++) to predict the fatigue performance of asphalt mixtures tested at test road project sites. In this VECD-FEP++ approach, the viscoelastic nature of asphalt concrete (AC) mixtures with growing damage is addressed using the VECD model, whereas the finite element program (FEP++) accounts for other important characteristics, such as temperature, layer thickness, stiffness gradient, etc. Comparisons between the field fatigue performance of the test road pavements to those predicted by the VECD-FEP++ simulations showed a generally positive relationship between model predictions and field observations. For a quick fatigue assessment, Underwood et al. [33,35,38] implemented the simplified VECD (S-VECD) model in the FEP++, instead of the original VECD model. The fatigue life of the pavements predicted using the S-VECD-FEP++ was found to agree well with the measured field response ($R^2$ = 0.8473% for the no-Terpolymer scenario, and $R^2$ = 0.9932 for the with-Terpolymer scenario). The S-VECD-FEP++ model was shown to be able to capture the effects of structure, climatic region, unbound layer modulus, and asphalt mixture properties, and to distinguish between top-down and bottom-up cracking.

The flexural bending test, also known as the beam fatigue test [50], is another testing method used to characterize the damage behavior of asphalt mixtures. This test measures the fatigue life of a compacted asphalt beam subjected to repeated flexural bending. According to De Mello et al. [94], what most likely happens in the field, and which is common to flexural fatigue tests, is a stress/strain field varying through the section from maximum compression to maximum tension in opposite sides instead of a homogeneous state of stress/strain throughout the sample section in cylindrical specimens subjected to uniaxial loading. The VECD approach was applied to flexural fatigue tests in studies with different goals, for example: (i) simplifying the calculation of damage parameters in the VECD model by considering the peak-to-peak values of stress and strain [95], (ii) evaluating which factors can influence the fatigue behavior more significantly [94], (iii) comparing fatigue cracking characteristics of a fine mix and a coarse mix [92], and (iv) studying the influence of RAP content on fine aggregate matrix (FAM) mixes [96]. Zhang et al. [96] also

concluded that the linear amplitude sweep (LAS) test of FAM mixes under flexural bending mode can provide acceptable data with good repeatability as an alternative test method for tests with cylindrical samples in the dynamic shear rheometer (DSR).

### 3.2. Fine Aggregate Matrix Approach

Microstructural discontinuities, such as air voids and microcracks, coalesce and propagate due to repeated dynamic loading from the traffic of heavy vehicles and environmental loads, giving rise to the fatigue cracking process. The fatigue cracking reduces the structural performance of the pavement, with a negative impact on its service life. This process develops under two circumstances: (i) after adhesive failure, when the crack occurs at the aggregate–mortar interface; and/or (ii) after cohesive failure, when the crack develops within the mortar. Based on such an interpretation of the cracking phenomenon, Kim et al. [8] began to study the fatigue behavior of asphalt mixtures using the fine aggregate matrix (FAM) approach, and developed a protocol to evaluate the FAM's properties.

The FAM is the matrix phase of the asphalt concrete composed of fine aggregates, filler, binder, and air voids, and represents the intermediate scale between the asphalt mastic and the full asphalt mixture. In studies with FAM mixtures, the primary assumption is that it reproduces the internal structure of the fine portion of the aggregate gradation of a full hot-mix asphalt (HMA) mixture. Another important assumption is that the physicochemical interactions between the aggregate and binder are replicated in the FAM specimen. Studies with a FAM gained notoriety for having a good agreement between the FAM and asphalt concrete (AC) properties, which was observed for the moisture characterization [97], fatigue cracking, and permanent deformation characterization [98–102].

Caro et al. [97] carried out surface energy measurements and dynamic mechanical analyzer tests on specimens of four FAM mixtures, and compared the results with the ones obtained by assessing the moisture susceptibility of the corresponding full asphalt mixtures by means of the saturation aging tensile stiffness (SATS) test [103]. A good agreement was observed between the results obtained for the fine-graded asphalt mixtures and the ones obtained for the dense asphalt mixtures. In both approaches, the granite mixture treated with 2% (by weight) hydrated lime was the most resistant to moisture damage, and the mixture containing only granite was the most susceptible to damage. However, the authors reported some differences between the FAM and HMA results for the sample containing crushed granite and limestone dust.

In order to investigate the inherent fatigue cracking resistance of modified asphalt binders (PPA, SBS, PPA + SBS, and PPA + Elvaloy), Motamed et al. [98] submitted FAM samples produced with modified asphalt binders and glass beads to torsional loading (controlled strain mode–275 kPa) at 10 Hz and 16 °C. The rationale for the use of glass beads in substitution for the mineral aggregate was to simulate the same stress state to which the binder was submitted in the asphalt concrete structure. The FAMs and the asphalt concrete mixtures fatigue lives (number of cycles to achieve 50% of the initial modulus) were compared, and it was observed that the FAM mixtures presented the same rank order for fatigue life of the asphalt concrete mixtures produced with the same modified asphalt binders [98].

Gudipudi and Underwood [18] observed a good agreement for the damage characteristic curves (C vs. S) between FAM and asphalt concrete for the tests carried out at 10 and 19 °C, but the C-values at failure for the FAMs were lower as compared to those of the asphalt concrete. It was not possible to compare results from tests with FAM and asphalt concrete carried out at 25 °C once the damage curve for both materials (FAM and asphalt concrete) presented a significant variation that could be related to viscoplasticity or another mechanism [18]. Coutinho [99] found the same rank order between the fatigue resistance of AC mixtures and the fatigue resistance of FAM mixtures when the FAM mixtures were subjected to stress-controlled time-sweep tests. Im et al. [100] observed a strong correlation between the linear and nonlinear viscoelastic and viscoplastic deformation characteristics of the asphalt concrete and its corresponding FAM. Underwood and Kim [104] evaluated

the effect of different compositions for the four material scales (binder, mastic, FAM, and asphalt concrete) using linear-viscoelastic properties, such as dynamic shear modulus ($|G^*|$) and phase angle ($\delta$). The authors concluded that the dynamic modulus and the phase angle for the FAM materials were much more similar to the full mixture data than were the mastic materials. The study showed that the materials at different scales presented differing levels of sensitivity to changes in the blending parameters. The materials at the FAM scale presented a sensitivity that was more in line with that observed for asphalt concrete mixtures under all of the tested conditions.

Palvadi et al. [17] validated the VECD theory to characterize damage in FAM specimens based on the similarity of the characteristic curves (CxS) for a given FAM for both monotonic and cyclic loading modes and different amplitudes. Palvadi et al. [17] also proposed a test procedure to investigate the healing characteristics of FAM specimens. This test procedure consists of four rest periods (5, 10, 20, and 40 min) and three levels of stiffness (20%, 30%, and 40% reduction in C). In this method, four specimens of each FAM were tested in order to apply a specific rest period in a specific specimen. Palvadi et al. [17] concluded that the healing percentage of each FAM is a material characteristic, once that the values for this parameter were similar, regardless of both the sequence of application of the rest period and the damage level. In an attempt to improve the procedure proposed by Palvadi et al. [17], Karki et al. [16,105] developed an integrated testing procedure that was capable of quantifying damage and healing characteristics using a single specimen without separating the damage and healing tests. Karki et al. [16,105] were the first to apply the simplified viscoelastic continuum damage (S-VECD) theory to characterize FAMs. The researchers highlighted that the characteristic curve (CxS) is a unique material property due to the similarity of the curves for a given material, regardless of the loading conditions (different amplitudes and frequencies) and the introduction of rest periods during the test.

Gudipudi and Underwood [18] analyzed the fundamental similarities or differences between FAM and AC scales by means of the S-VECD theory. They observed similarities in material properties between the two material scales, and the CxS curves for a particular FAM and its corresponding AC mix were very similar. However, the C-value when failure occurred was generally lower in the FAM as compared to the mix. This result suggested that the FAM can reach a greater damage accumulation before failure occurs. The use of FAM testing for material characterization and ranking of AC mixtures has a great potential, if the material fabrication protocols are accurately followed. Freire et al. [106] applied the S-VECD theory to evaluate the effect of different maximum nominal aggregate sizes (MNS) of the mineral aggregate particles on the FAM fatigue resistance to identify which one best represented the damage characteristics of the asphalt mixture. The FAM mixtures were prepared with three different MNS (4.00, 2.00, and 1.18 mm), and their fatigue characteristics were evaluated and compared to the ones of a hot-mix asphalt (HMA) mixture prepared with an MNS of 12.5 mm. The main finding was that the Wöhler curves created for the FAMs produced with mineral aggregate particles of 2.00 mm and the corresponding asphalt mixture presented similar trends. The authors pointed out that a direct comparison of the absolute results, instead of the trends observed, could not be done directly because the parameters for the HMA mixture were obtained by axial loading, whereas the FAM parameters were obtained by shear loading.

Some researchers adapted the linear amplitude sweep test (LAS) method proposed by Johnson [107] to characterize the fatigue resistance of the FAMs using the VECD approach. The investigations evaluated the effect of (i) different particle size distributions [99], (ii) different nominal maximum aggregate size [108,109], (iii) the thermal and photochemical aging [110,111], and (iv) the effect of RAP content and rejuvenating agents (RAs) [112,113]. However, Freire et al. [109] did not recommend the use of the LAS test to analyze the fatigue resistance of the FAM mixes due to the difficulty to achieve the failure, once the torque capacity of the DSR was low and unable to take the sample to failure. The authors observed that for the highest strain amplitudes of the LAS test, the equipment needed to work near its capacity due to the high stiffness of the FAM specimens.

Regarding the FAM mixes containing RAP and recycled asphalt shingle (RAS), researchers [6,73,96,101,112–114] concluded that the use of these materials decreased the fatigue life of the mixture due to the hard binder present in the RAS and RAP. The use of RAs (petroleum tech, green tech, and agriculture tech) in the FAM mixes containing RAS and RAP was investigated by Zhang et al. [96], Nabizadeh [101], Nabizadeh et al. [114], and Zhu et al. [6] as an alternative to increase the fatigue life of the FAMs. Nabizadeh [101] and Zhang et al. [96] concluded that the RAs resulted in softer mixtures with improved fatigue life (especially for the FAMs with high RAP contents). Zhu et al. [6] observed the same behavior in the case of the FAM with RAS mixed with another petroleum-based RA. The combination of the WMA additive with the petroleum tech rejuvenator was evaluated by Nabizadeh [101], and this combination resulted in the softest FAM compared with other rejuvenators (green tech and agriculture tech).

With the aim of investigating the fatigue cracking of the asphalt binders at the FAM scale without the physicochemical interaction with the mineral aggregate, Motamed et al. [98] used rigid particles, such as glass beads, in substitution for the mineral aggregate to produce the FAM specimens. This new technique resulted in similar fatigue cracking characteristics between the FAMs and the asphalt mixtures produced with the same asphalt binder. The researchers concluded that the glass beads could be used in substitution for the mineral aggregate when the binder properties were the main issue of interest. More recently, Li et al. [115] used a combined fatigue–healing method based on the VECD model to evaluate fatigue and self-healing properties of three rock asphalt composites. Li et al. [115] indicated that the replacement of a portion of the virgin asphalt binder by the rock asphalts enhanced the fatigue cracking resistance of the FAM mixtures, and the influence on fatigue life was dependent on both the type and concentration of the rock asphalt.

Warm fine aggregate mixtures (W-FAM) fabricated using different WMA additives were compared with an HMA (control mixture) in a study by Sadeq et al. [116]. The control mixture presented higher dissipated pseudo-strain energy (DPSE) than the W-FAM, and the fatigue life in the VECD analysis approach was not statistically significant among the control mixture and the W-FAM mixtures, indicating that the WMA mixtures had fatigue resistance comparable to the hot-mix asphalt mixtures. In studies by Sadek et al. [117] and Sharma and Swamy [118], a probabilistic analysis approach was applied to the fatigue life prediction model deduced from the VECD theory. The inherent variability of asphaltic materials exhibited in the fatigue test results led the researchers to develop a new probabilistic approach. Probabilistic approaches present the ability to account for uncertainties associated with fatigue tests, models, and model parameters. In a study conducted by Sadek et al. [119], the efficacy of using the probabilistic approach in the analysis of the viscoelastic continuum damage (VECD) and fatigue life was examined for hot and warm fine aggregate mixtures (H-FAM and W-FAM). The probabilistic VECD approach had the advantage of providing more reliable predictions of fatigue life that accounted for uncertainty in determining the model parameters, instead of the deterministic approach. The probabilistic analysis results showed that the W-FAMs had shorter fatigue lives than the one obtained by the control H-FAM mixture; however, their fatigue lives presented more consistency and less uncertainty.

## 4. Analysis Protocol of Tests with FAM Using the S-VECD Approach

The outputs of tests carried out using fine aggregate matrix (FAM) specimens can be analyzed by means of the theory of continuum mechanics, by performing a linear-viscoelastic characterization followed by a damage evolution characterization. Table 1 presents the resulting data obtained from fingerprint and damage tests, along with a summary of the equations employed to determine the linear-viscoelastic and damage properties, and Table 2 presents the equations employed to build the FAM prediction fatigue models.

**Table 1.** Summary of the VECD theory applied to asphalt mixtures—fingerprint and damage tests.

| Fingerprint Test | | | FAM1 Sample 1 |
|---|---|---|---|
| 1. Data obtained from the fingerprint | Dynamic shear modulus within the linear-viscoelastic region ($\|G^*_{LVE}\|$), relaxation rate (m) (Figure 3a,b) | - | $\|G^*\|_{LVE} = 9.93 \times 10^8$<br>m = 0.476 |
| 2. Prony series fitted to the storage modulus values to obtain $G_e$, $\rho_i$, and $G_i$. | $G'(\omega) = G_e + \sum\limits_{i=1}^{n} \dfrac{G_i \omega^2 \rho_i^2}{\omega^2 \rho_i^2 + 1}$ | Equation (56) | $G_e$, $\rho_i$, and $G_i$ are calculated by using Solver |
| 3. Laplace transform to convert data from frequency to time domain | $G(t) = G_\infty + \sum\limits_{i=1}^{n} G_i e^{-t/\rho_i}$ | Equation (58) | G(t) values obtained by using the parameter of the Prony series |
| 4. Model adjusted to data to obtain the material parameter, m | $G(t) = G_0 + G_1 t^{-m}$ (Figure 3b) | Equation (59) | $G_0 = 1$<br>$G_1 = 2.65 \times 10^8$<br>m = 0.476 |
| 5. Equation to obtain the material parameter, $\alpha$ | $\alpha = (1 + 1/m)$ for strain-controlled tests<br>$\alpha = 1/m$ for stress-controlled tests | Equation (30)<br>Equation (31) | $\alpha = 2.10$ |
| Damage Test | | | FAM1 sample 1 k = 5 |
| 1. Data obtained from the damage tests | Complex modulus ($\|G^*\|$), phase angle ($\varphi$), and strain ($\varepsilon$), at each cycle k | - | $\|G^*\|_{k=1} = 7.87 \times 10^8$<br>$\|G^*\|_{k=5} = 6.91 \times 10^8$<br>$\varphi_{k=5} = 49.5$<br>$\varepsilon_{k=5} = 0.051\%$ |
| 2. Peak pseudo strain during each cycle k, $\varepsilon_m^R(t)$ | $\varepsilon_0^R(t) = \varepsilon_0 \|G^*\|_{LVE}$ | Equation (45) | $\varepsilon_0^R(t) = \varepsilon_{0k=5} = 5.03 \times 10^5$ |
| 3. Pseudo stress, $\sigma^R$ | $\sigma^R = \sigma$ | Elastic-viscoelastic correspondence principle | $\sigma^R = 350$ kPa |
| 4. Initial pseudo stiffness, I | $I = \dfrac{\|G^*\|_{k=1}}{\|G^*\|_{LVE}}$ | Dynamic shear modulus at the first cycle | $I = \dfrac{7.87 \times 10^8}{9.93 \times 10^8} = 0.792$ |
| 5. Normalized pseudo stiffness at each cycle k, C | $C_k(S) = \dfrac{\|G^*\|_{k=n}}{I\|G^*\|_{LVE}}$ | Equation (47) | $C_5 = \dfrac{6.91}{0.792 \times 7.87} = 0.88$ |
| 6. Damage parameter, S | $dS \equiv \left[\frac{1}{2}\left(\varepsilon_{0,k,PP}^R\right)^2 (C_{k-1} - C_k)\right]^{\alpha/(1+\alpha)} (t_k - t_{k-1})^{1/(1+\alpha)}$ | Equation (48) | $S_5 \equiv 4.71 \times 10^7$ |
| 7. Characteristic curve, CxS | The characteristic curves, CxS, are obtained by cross-plotting the C values against the S values at each cycle k | Figure 4a | $C_5 = 0.88$ vs. $S_5 = 4.71 \times 10^7$ at cycle k = 5 |

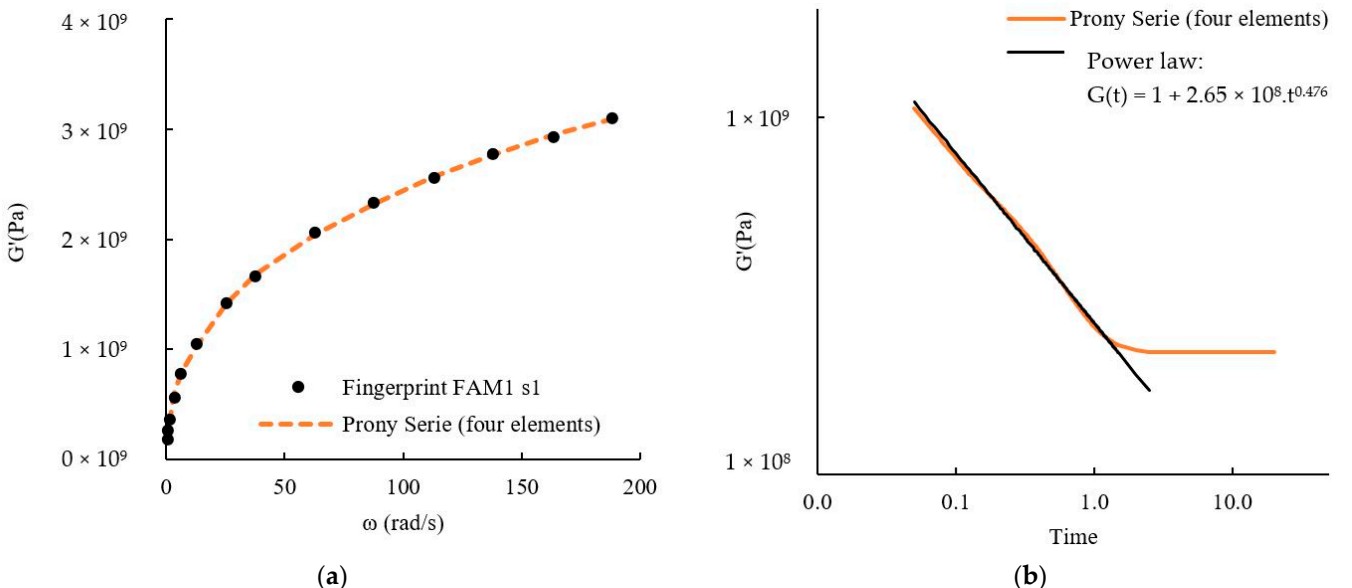

**Figure 3.** Fingerprint: (**a**) relaxation modulus vs. frequency; and (**b**) relaxation modulus vs. time (log).

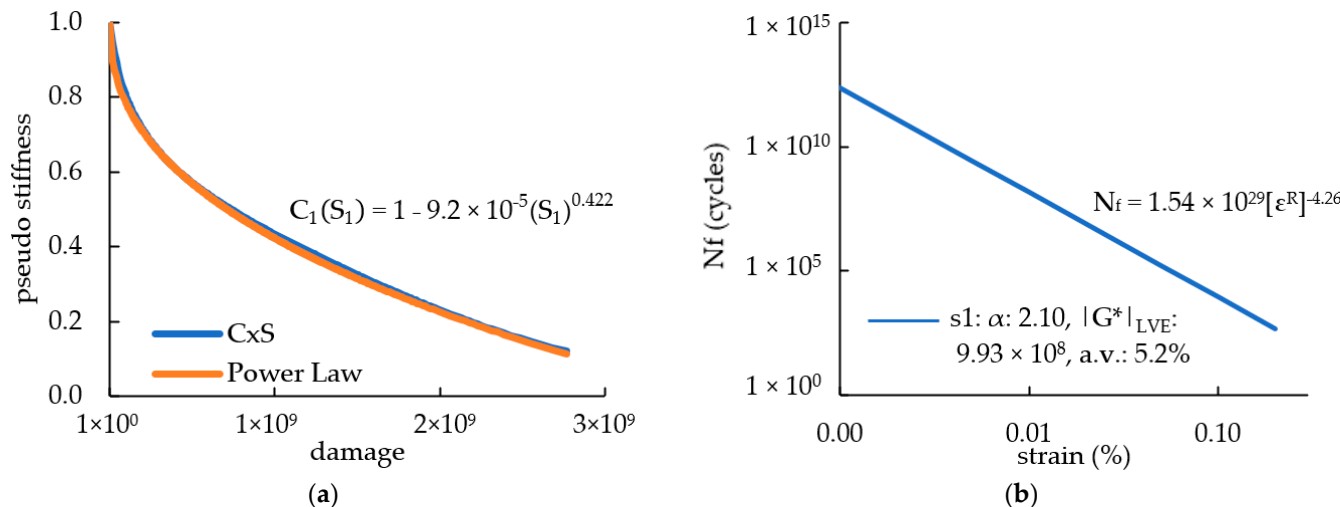

**Figure 4.** (**a**) CxS curve of the specimen 1-FAM1 and power law fitted to data; and (**b**) fatigue model for the specimen 1-FAM1.

**Table 2.** Summary of the VECD theory applied to asphalt mixtures—fatigue life prediction.

| Fatigue Life Prediction Model | | | |
|---|---|---|---|
| 1. Power law fitted to the CxS curve to obtain $C_{10}$, $C_{11}$, and $C_{12}$ | $C_1(S_1) = C_{10} - C_{11}(S_1)^{C_{12}}$ | Equation (40) | Figure 4a |
| 2. Calculation of the parameter A | $A = f\left\{\frac{1}{2}C_1C_2\right\}^{\alpha}\{1 + \alpha(1 - C_2)\}^{-1}S_f^{[1+\alpha(1-C_2)]}$ | Equation (49) | $A = 7.35 \times 10^{29}$ |
| 3. Calculation of the parameter B | $B = 2\alpha$ | Equation (50) | $B = 4.35$ |
| 4. Prediction fatigue life curve | $N_f = A[\varepsilon^R]^{-B}$ | Equation (51) | Figure 4b |
| 5. $N_f$ (strain = 0.005%) | $N_{f(0.005\%)} = 7.35 \times 10^{29}[4.46 \times 10^4]^{-4.35}$ | Equation (51) | $4.59 \times 10^9$ |
| 6. $N_f$ (strain = 0.2%) | $N_{f(0.2\%)} = 7.35 \times 10^{29}[1.78 \times 10^6]^{-4.35}$ | Equation (51) | 500.25 |

## 5. Application of the S-VECD Theory: Laboratory Tests and Discussion of Results

### 5.1. Experimental Method—Materials and Preparation of the FAM Specimens

The effect of the addition of different proportions of reclaimed asphalt pavements (RAP) and virgin binder on the fatigue performance of fine aggregate matrices was evaluated by performing tests according to the protocol proposed by Kim and Little [5], and by analyzing the obtained data applying the simplified viscoelastic continuum damage (S-VECD) theory, following the procedures described in Tables 1 and 2. Fingerprint and damage tests under the stress-controlled mode of loading were carried out using cylindrical fine aggregate matrix (FAM) specimens (40 mm in height and 12 mm in diameter). The FAMs were produced with two RAP proportions (20% and 40%). The binder content was adjusted by using either a performance grade (PG) 58-16 binder or a PG 64-22 binder. One source of virgin mineral aggregate was used—a basalt rock obtained from Bandeirantes Quarry, located in the city of Sao Carlos, state of Sao Paulo, Brazil. One RAP source obtained from roads around the city of Sao Carlos was used in the study. The mineral aggregate and asphalt binder were characterized following the standard procedures from the American Society for Testing and Materials (ASTM D7928, ASTM C128, ASTM D70, ASTM D6373, ASTM D6648, and ASTM D7175) and from the American Association of State Highway and Transportation Officials (AASHTO T84, AASHTO T85, and AASHTO T209). The mineral aggregate and asphalt binder characteristics are presented in Table 3. Aggregate particles passing sieve #10 (2.00 mm) were used. The FAM aggregate gradation met the specification of the Brazilian Department of Terrestrial Infrastructure (DNIT 031/2004-ES). A dense-graded aggregate gradation represented by the middle of the C band was chosen, as it is a typical mineral composition used for road construction in Brazil. The RAP binder content was 8.3%. The binder content to be used to cover the virgin fine aggregate particles

was estimated by means of the specific surface method [120], and the result was 9.0%. The difference between the actual binder content and the binder content calculated by means of the specific surface method was compensated by adding a complementary proportion of virgin asphalt binder (PG 58-16 or PG 64-22).

**Table 3.** Material characteristics.

| **Basalt Rock** | | | |
|---|---|---|---|
| Quarry identification | Bandeirantes | | |
| Specific gravity of coarse aggregates (g/cm$^3$) | 2.904 | | AASHTO T 85 |
| Specific gravity of fine aggregates (g/cm$^3$) | 2.999 | | AASHTO T 84 |
| Specific gravity of filler (g/cm$^3$) | 2.769 | | ASTM D7928 |
| Absorption (%) | 0.6 | | ASTM C128 |
| **RAP Material** | | | |
| Quarry location | São Carlos/SP | | |
| Maximum specific gravity (g/cm$^3$) | 2.596 | | AASHTO T209 |
| **Asphalt Binders** | | | |
| Performance grade (PG) | PG 58-16 | PG 64-22 | ASTM D6373 |
| Specific gravity (g/dm$^3$) | 1.015 | 1.004 | ASTM D70 |
| Continuous grade—virgin (°C) | 61.07 | 66.84 | ASTM D7175 |
| Continuous grade—short-term aged (°C) | 65.52 | 66.94 | ASTM D7175 |
| Continuous grade—long-term aged (S [60][1]) | −20.7 | −26.9 | ASTM D6648 |
| Continuous grade—long-term aged (m [60][2]) | −20.2 | −27.4 | ASTM D6648 |

[1] Creep stiffness at 60 s (MPa); [2] slope at 60 s.

The results presented here refer to the following FAMs: (i) FAM1: 20% of RAP and binder PG 64-22; (ii) FAM2: 40% of RAP and binder PG 64-22; (iii) FAM3: 20% of RAP and binder PG 58-16; and (iv) FAM4: 40% of RAP and binder PG 58-16. The mixtures were compacted in a specially designed mold with internal dimensions of 40 mm in length and 12.8 mm in diameter in order to produce the cylindrical FAM specimens.

*5.2. Experimental Method: Fingerprint Test*

Fingerprint tests were carried out in an Anton Paar MCR 302 dynamic shear rheometer (DSR). The linear-viscoelastic properties of the FAMs were obtained by performing a fingerprint test at a stress of 15 kPa. This stress was selected within the linear-viscoelastic (LVE) range in order to avoid deformations larger than 100 μstrain at the end of the test. The frequencies were the following: 30, 26, 22, 18, 14, 10, 6, 4, 2, 1, 0.5, 0.2, 0.1, 0.05, and 0.01 (Hz). The software used to control the DSR was the Rheoplus/32 V3.62, and the data were exported to a Microsoft Excel spreadsheet. A Prony series (Equation (55)) was fitted to the storage modulus $G'(\omega)$ values obtained in the fingerprint test by using the Solver tool in Microsoft Excel in order to obtain the equation parameters. The static relaxation shear modulus as a function of time, $G(t)$, was predicted by applying a Laplace transform (Equation (57)), which employed the Prony series parameters previously obtained. For the sake of exemplification, Figure 3a shows the results of $G'(\omega)$ versus frequency ($\omega$) obtained for the specimen 1-FAM1, and Figure 3b shows a log–log chart of the $G(t)$ values, where the slope of the log–log $G(t)$ curve (Equation (32)) was obtained by adjusting a power law function (Equation (58)). The material damage evolution rate, $\alpha$, was calculated according to Equation (31), as the damage tests were conducted under controlled stress [29]. The linear-viscoelastic complex modulus ($|G^*|_{LVE}$) was the averaged $|G^*|$ values measured at 1 Hz in the fingerprint.

*5.3. Experimental Method: Damage Test*

Damage tests were carried out in the Anton Paar MCR 302 dynamic shear rheometer (DSR). Oscillatory tests under controlled stress at a frequency of 1 Hz and a temperature of 25 °C were performed in order to evaluate the damage evolution process of the FAM

specimens. The damage tests were performed by using the same sample tested in the fingerprint test, except for the cases in which the specimen presented deformations larger than 100 μstrain at the last frequency of the test. In these cases, those specimens were discarded and new specimens were tested. The applied stress adopted to damage the specimens was 350 kPa. The data were exported from Rheoplus/32 V3.62 software to a Microsoft Excel spreadsheet. The pseudo stiffness (C) and the accumulated damage (S) values were calculated by means of the equations of the S-VECD theory (Equations (46)–(48)). The characteristic curves, CxS, were built according to the model proposed by Lee and Kim [25]. The material fatigue life, $N_f$, which is the number of axle load repetitions capable of leading the material to failure, was predicted by employing the mechanistic fatigue life prediction model (Equation (51)) developed by Lee et al. [7] and Kim and Little [5]. The fatigue models were adjusted for a 50% reduction in the material's pseudo stiffness. Figure 4a,b shows an example of the CxS curve and the fatigue curve obtained for a single specimen. The detailed description of the procedure devised to deal with replicates of the same FAM and to build the CxS and fatigue curves is presented in the next section. Figure 4a presents the CxS curve built from laboratory data and the power law fitted to the data for the specimen 1-FAM1. Figure 4b presents the fatigue curve for the specimen 1-FAM1.

### 5.4. Analysis and Discussion of Results

In general, distinct specimens of the same FAM can present different values for the linear-viscoelastic and damage properties. The heterogeneity in the material properties for specimens of the same material may result in different characteristic curves. In order to overcome this issue and to accurately predict the damage behavior of the mixtures, the average properties of at least three specimens was considered in the model. By doing so, it was possible to obtain similar curves for distinct specimens of the same FAM. After averaging the properties and plotting the material CxS curves, an average CxS curve that best represented the material damage behaviour was built for each FAM. Table 4 presents the air voids and the linear-viscoelastic properties (linear-viscoelastic complex modulus, $|G^*|_{LVE}$, and the material relaxation rate, m) for each FAM specimen. Table 4 also shows the average linear-viscoelastic properties of each FAM and the coefficient of variation of each specimen. The specimen shape factor and the $D^R$ criterion were tools that were able to check the specimen-to-specimen variability, and they are also presented in Table 4. As recommended by Lee et al. [7], the specimen-to-specimen variability can be quantified and minimized by means of a shape factor. By following such an idea, a shape factor was defined in this study as the ratio of the area above a particular specimen's CxS curve to the area above the average CxS curve of the FAM. A shape factor close to 1 indicated a valid specimen; i.e., the results of a particular specimen were very close to the FAM average curve. In the study by Lee at al. [7], replicates with shape factors greater than 1.1 or lower than 0.9 were considered outliers. By following this premise, curves with shape factors of $1.0 \pm 0.1$ were accepted and used to build the characteristic curve of the FAMs. In the calculation of the shape factors, a decay until a pseudo stiffness equal to 0.3 was considered. The shape factors of all specimens are presented in Table 4, where it is possible to observe shape factors between 0.92 and 1.08, except for FAM2 s1, which presented a shape factor of 0.8. The other tool applied in this study to check the specimen-to-specimen variability was the $D^R$ criterion, which is defined as the average reduction in pseudo stiffness up to failure [48]. Wang and Kim [48] found a variation of $\pm 0.04$ in the $D^R$ value of specimens of the same mixture. The authors also carried out sensitivity studies of the pavement performance analysis using the S-VECD model with the $D^R$ failure criterion, and they found that with this variation of $\pm 0.04$ in the $D^R$ value, the predicted fatigue damage for the pavements did not differ significantly. The variation between the specimens of the FAMs evaluated in this study was $\pm 0.05$.

**Table 4.** Linear-viscoelastic properties and fatigue model parameters of the FAMs.

| Material | Sample | Air Voids | Linear-Viscoelastic Properties | | | | | | CxS Parameters | |
|---|---|---|---|---|---|---|---|---|---|---|
| | | | m | $M_{(av)}$ | cv (%) | $|G^*|_{LVE}$ (kPa) | $|G^*|_{LVE\,(av)}$ (kPa) | cv (%) | Shape Factor | $D^R$ |
| FAM1 | s1 | 5.15 | 0.476 | | 3.2 | $9.93 \times 10^8$ | | 11.3 | 1.05 | 0.382 |
| | s2 | 5.28 | 0.490 | | 6.2 | $7.53 \times 10^8$ | | −15.6 | 0.92 | 0.373 |
| | s3 | 4.96 | 0.422 | 0.461 | −8.6 | $9.46 \times 10^8$ | $8.92 \times 10^8$ | 6.1 | 0.97 | 0.350 |
| | s4 | 4.99 | 0.467 | | 1.2 | $8.82 \times 10^8$ | | −1.1 | 0.94 | 0.345 |
| | s5 | 4.99 | 0.452 | | −2.0 | $8.85 \times 10^8$ | | −0.8 | 1.08 | 0.394 |
| FAM2 | s1 | 5.19 | 0.342 | | −8.7 | $1.57 \times 10^9$ | | −7.6 | 0.8 | - |
| | s2 | 5.05 | 0.386 | 0.375 | 2.9 | $1.61 \times 10^9$ | $1.70 \times 10^9$ | −5.3 | 0.99 | 0.332 |
| | s3 | 5.09 | 0.380 | | 1.3 | $1.96 \times 10^8$ | | 15.3 | 1.03 | 0.332 |
| | s4 | 5.05 | 0.392 | | 4.5 | $1.66 \times 10^9$ | | −2.4 | 0.98 | 0.325 |
| FAM3 | s1 | 5.16 | 0.463 | | 0.7 | $5.04 \times 10^8$ | | −15.3 | 1.04 | 0.395 |
| | s2 | 4.67 | 0.465 | 0.460 | 1.1 | $5.90 \times 10^8$ | $5.95 \times 10^8$ | −0.8 | 0.96 | 0.415 |
| | s3 | 5.47 | 0.448 | | −2.5 | $6.61 \times 10^8$ | | 11.1 | 0.99 | 0.406 |
| | s4 | 5.09 | 0.463 | | 0.7 | $6.24 \times 10^8$ | | 4.9 | 0.91 | 0.360 |
| FAM4 | s1 | 4.57 | 0.398 | | −1.8 | $1.54 \times 10^9$ | | −2.1 | 1.03 | 0.455 |
| | s2 | 4.68 | 0.402 | 0.406 | −1.0 | $1.63 \times 10^9$ | $1.57 \times 10^9$ | 3.6 | 1.00 | 0.448 |
| | s3 | 4.89 | 0.417 | | 2.7 | $1.55 \times 10^9$ | | −1.5 | 0.97 | 0.423 |

Regarding the effect of the RAP content, the linear-viscoelastic properties in Table 4 show that the FAMs containing 40% of RAP (2 and 4) presented higher $|G^*|_{LVE}$ values as compared to the mixtures containing 20% of RAP (1 and 3). For the same RAP proportion (20% or 40%), the highest complex modulus was observed for the FAMs containing the binder PG 64-22. Regarding the relaxation rate, m, the increase in the RAP content from 20 to 40% led to a decrease in this rate, with FAMs 1 and 3 (20% of RAP) presenting higher m values as compared to FAMs 2 and 4 (40% of RAP). Among the FAMs prepared with 20% of RAP, FAM1 (prepared with binder PG 64-22) and FAM3 (prepared with binder PG 58-16) presented equivalent m values (FAM3 presented a slight 0.2% reduction as compared to the FAM1 rate). Among the FAMs prepared with 40% of RAP, FAM4 (prepared with binder PG 58-16) presented an increase of 8.5% in the m value as compared to FAM2 (prepared with binder PG 64-22).

Figure 5 illustrates the CxS curves for each specimen built by using its own properties, while Figure 6 shows the CxS curves for each specimen built by using the average linear-viscoelastic properties. When comparing Figure 5 to Figure 6, it is possible to observe that the CxS curves tended to converge when the average properties were employed. However, this procedure did not yet result in a single CxS curve for each material. For such a reason, an average CxS curve for each FAM was built, which represented the material damage behavior, by using the Solver tool in Microsoft Excel. These curves are illustrated in Figure 6. It can be observed that FAM2 s1 is not presented in Figure 6, and this was due to its shape factor result of 0.8, which was lower than the limit of 0.9. For such a reason, the specimen FAM2 s1 was not included in the specimen set used to build the average characteristic curve of FAM2.

The FAM fatigue models (Figure 7b) were adjusted to a damage level (S) corresponding to a reduction of 50% in the material pseudo stiffness (C). These S values were obtained from the final (average) CxS curves (Figure 7a). By means of the fatigue models, it was possible to compare the effect of the different proportions of RAP (20% and 40%) and the effect of the addition of the asphalt binders (PG 64-22 and PG 58-16). The fatigue curves (Figure 7b) indicate that the materials behaved differently at different strain levels. For this reason, two strain levels were adopted for this analysis (0.005% and 0.20%). The parameters A and B of the fatigue models are presented in Table 5, along with the fatigue lives for each strain level and the mixture ordering (with the first positions occupied by the materials with the longest fatigue lives). Another procedure that can be applied to rank the FAM fatigue lives is the mixture fatigue factor (MFF) proposed by Nascimento et al. [121]. The

MFF is determined by calculating the area below the curve "number of cycles vs strain" between two specific strains. Table 5 also presents the FAM fatigue factors ($FF_{FAM}$) and the rank order of the FAMs according to this factor, by assuming that the higher the factor, the higher the fatigue performance of the material.

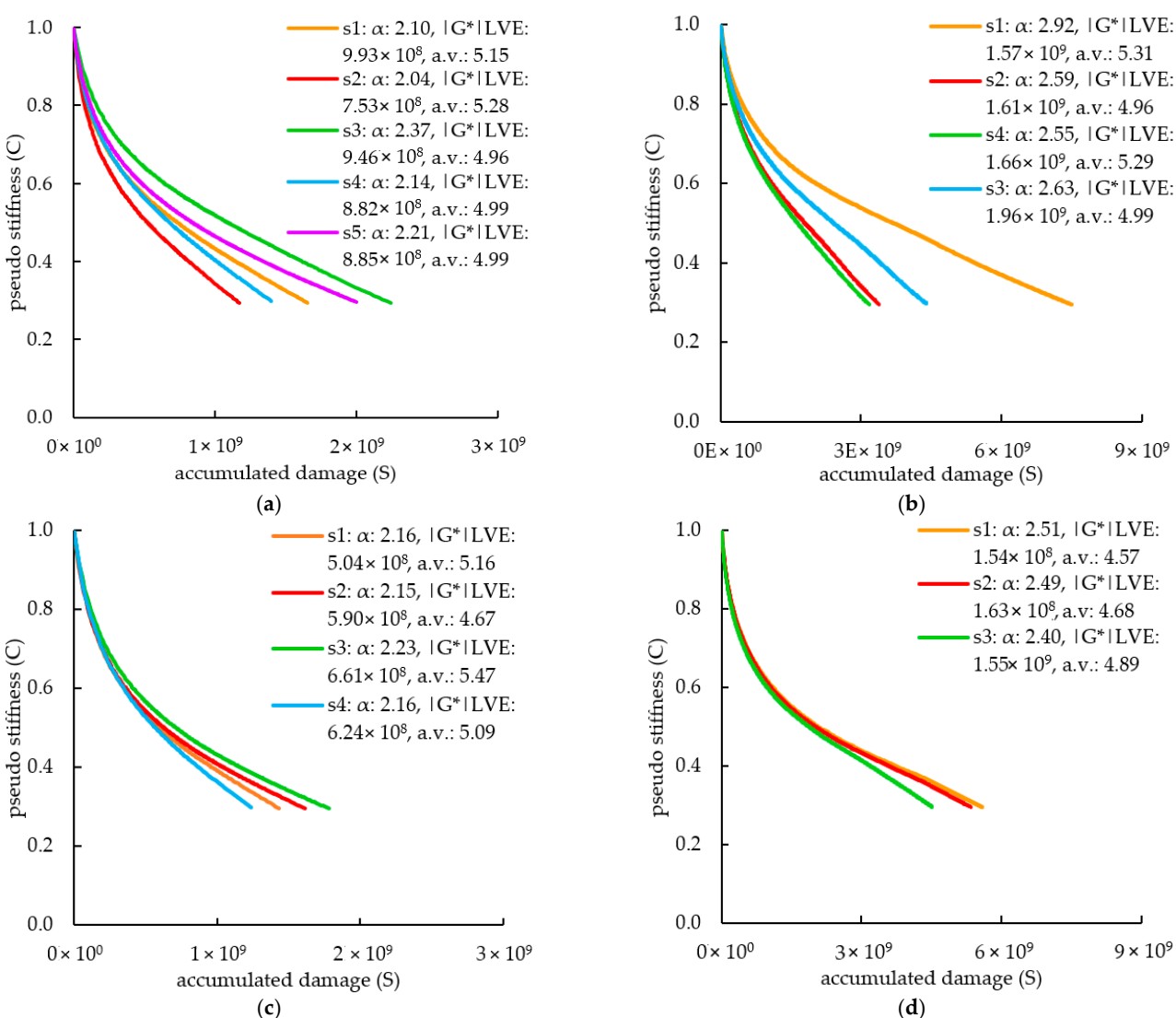

**Figure 5.** Individual CxS curves obtained for each specimen. (**a**) FAM1, (**b**) FAM2, (**c**) FAM3, (**d**) FAM4.

FAMs 1, 2, 3, and 4 were compared in order to evaluate the effect of the addition of different proportions of RAP (20% and 40%) to the mixture. FAMs 2 and 4, containing 40% RAP, present higher stiffness as compared to FAMs 1 and 3, which contained 20% RAP, as well as lower relaxation rates, higher damage evolution rates, and higher parameters A and B of the fatigue model (Tables 4 and 5). By comparing FAMs 1 and 2, it was possible to evaluate the effect of the increase in the RAP content from 20% to 40%, when a binder PG 64-22 was added to adjust the binder content of the RAP material. FAM2 (40% RAP) presented a higher stiffness and higher damage evolution rate than FAM2 (20% RAP). At low strains, FAM2 presented an increase of 2.6 times in its fatigue life when compared to FAM1; however, at high strains, the fatigue life of FAM2 was about 6.4% of the fatigue life of FAM1.

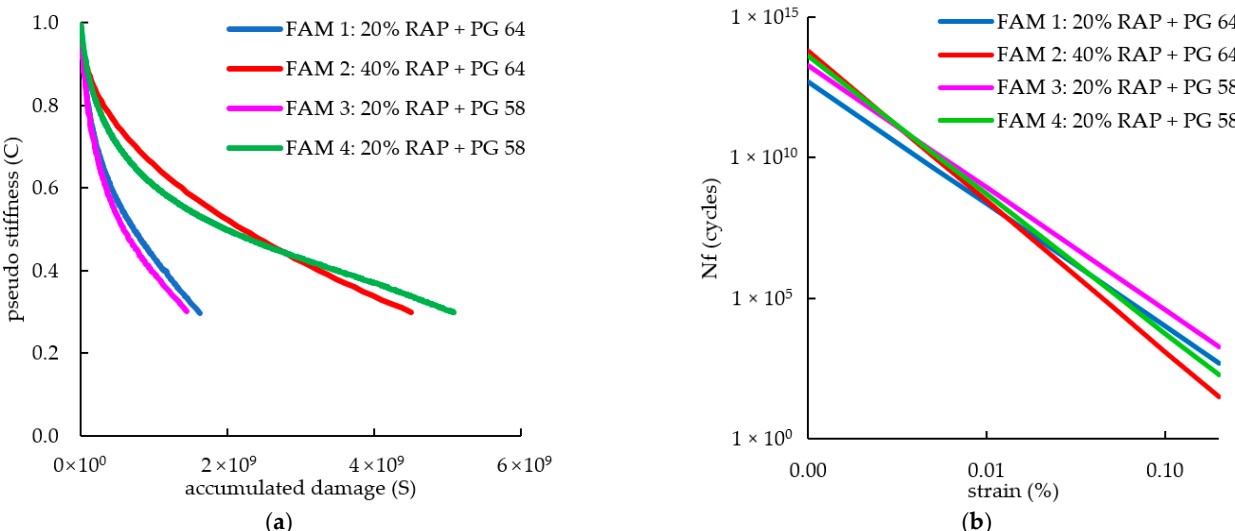

**Figure 6.** Individual CxS curves obtained by using the average viscoelastic properties, and average CxS curves. (**a**) FAM1, (**b**) FAM2, (**c**) FAM3, (**d**) FAM4.

**Figure 7.** (**a**) Resulting CxS curves; and (**b**) resulting fatigue models.

**Table 5.** Parameters A and B of the fatigue models and FAM fatigue factors.

| FAM | A | B | $N_f$ (0.005%) | Rank Order | $N_f$ (0.20%) | Rank Order | $FF_{FAM}$ | Rank Order |
|---|---|---|---|---|---|---|---|---|
| FAM1 | $7.35 \times 10^{29}$ | 4.35 | $4.59 \times 10^9$ | 4 | 500.25 | 2 | 2.32 | 3 |
| FAM2 | $2.68 \times 10^{36}$ | 5.35 | $1.17 \times 10^{10}$ | 3 | 31.80 | 4 | 2.30 | 4 |
| FAM3 | $5.25 \times 10^{29}$ | 4.35 | $1.77 \times 10^{10}$ | 1 | 1885.85 | 1 | 2.49 | 1 |
| FAM4 | $2.16 \times 10^{34}$ | 4.93 | $1.52 \times 10^{10}$ | 2 | 189.63 | 3 | 2.39 | 2 |

By comparing FAMs 3 and 4, it was possible to evaluate the effect of the increase in the RAP content from 20% to 40% when a binder PG 58-16 was added to adjust the binder content of the RAP material. FAM4 (40% RAP) presented a higher stiffness and higher damage evolution rate than FAM3 (20% RAP). At low strains, FAM4 presented a decrease of 14.5% in its fatigue life when compared to FAM3, and at high strains, the fatigue life of FAM4 was about 10% of the fatigue life of FAM3. In summary, the increase in the RAP content from 20% to 40% increased the fatigue life at low strains only if the binder content was corrected with a binder PG 64-22 and reduced it at high strains.

FAMs 1, 2, 3, and 4 were compared in order to evaluate the effect of adding the binder PG 64-22 or the binder PG 58-16 to adjust the binder content of the RAP material. When the FAMs containing 20% of RAP (FAMs 1 and 3) were compared, the results indicated a stiffness 33% lower for FAM3 (RAP binder content adjusted with the binder PG 58-16), and a similar damage evolution rate for both FAMs 1 and 3. At low strains, FAM3 presented a fatigue life 3.9 times longer than FAM1 (RAP binder content corrected with the binder 64-22), and at high strains, the fatigue life of FAM3 was 3.77 times longer than that of FAM1. When the FAMs containing 40% of RAP (FAMs 2 and 4) were compared, it is possible to observe that the stiffness and the damage evolution rate of FAM2 (RAP binder content adjusted with the binder PG 64-22) were about 8% higher than the values for FAM4 (RAP binder content adjusted with the binder PG 58-16). At low strains, FAM2 presented a fatigue life that was 17% of the one for FAM4, and at high strains, FAM4 has a fatigue life that was 1.3 times longer than the one expected for FAM2. In summary, the use of the binder PG 58-16 to adjust the FAM binder content increased the fatigue life at both low and high strains.

The rank order at low strains showed that FAM3, prepared with 20% RAP and the binder content of the RAP material adjusted with the binder PG 58-16, presented the longest fatigue life; and FAM1, prepared with 20% RAP and the binder content of the RAP material adjusted with the binder PG 64-22, presented the shortest fatigue life. FAM4, prepared with 40% RAP and the binder content of the RAP material adjusted with a binder PG 58-16, occupied the second position in the rank order, followed by FAM2, prepared with 40% RAP and the binder content of the RAP material adjusted with a binder PG 64-22, occupying the third position. The rank order at high strains showed FAM3 occupying the first position, followed by FAMs 1, 4, and 2. Concerning the fatigue factors (FFFAM), the rank order showed that FAMs 3 and 4, prepared with 20% and 40% of RAP, respectively, and the binder content of the RAP material adjusted with the binder PG 58-16, occupied the first and second positions, respectively. FAMs 1 and 2, prepared with 20% and 40% of RAP, respectively, and the binder content of the RAP material adjusted with the binder PG 64-22, occupied the third and fourth positions, respectively.

Concerning the $D^R$ values presented in Table 4, it can be observed that FAM4, prepared with 40% RAP and the binder content of the RAP material adjusted with the binder PG 58-16, presented the higher $D^R$ values, followed by FAM3, prepared with 20% RAP and the binder content of the RAP material adjusted with the binder PG 58-16. It can also be observed that the lower values of the $D^R$ parameter were the ones of FAM2, which was prepared with 40% RAP and the binder content of the RAP material adjusted with the binder PG 64-22. The trend observed by Wang and Kim (2019) was that the $D^R$ value decreased as the RAP content increased, which was the same trend observed in this study

for the two mixtures prepared with the binder PG 64-22. However, for the two mixtures prepared with the binder PG 58-16, the $D^R$ values increased with the increase in RAP. Wang and Kim (2019) emphasized that the $D^R$ value alone cannot be used to compare the fatigue performance of different asphalt mixtures.

The overall findings indicated that: (i) the increase in RAP from 20% to 40% increased the fatigue life of the mixture prepared with the binder PG 64-22, decreased the fatigue life of the mixture prepared with the binder PG 58-16 at low strains, and decreased the fatigue life of the mixtures at high strains; (ii) the addition of the binder PG 58-16 instead of the binder PG 64-22 resulted in an increase in the mixture fatigue life at both low and high strain levels; and (iii) the rank order of the FAM fatigue factors indicated that the increase in RAP from 20% to 40% decreased the fatigue life of the mixtures. Figure 8 depicts the experimental method followed in this study.

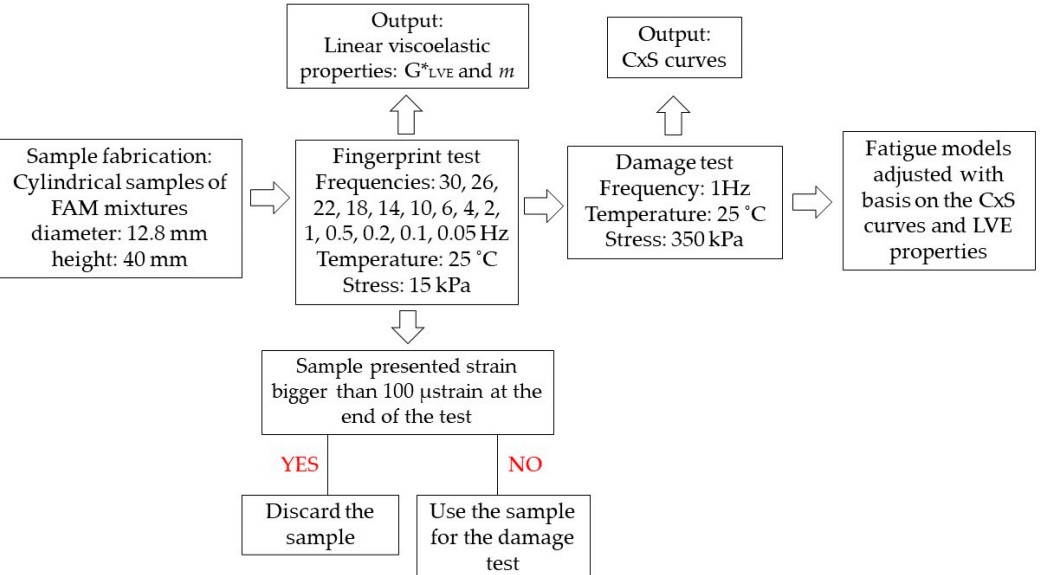

**Figure 8.** Experimental method followed in this study.

## 6. Conclusions

This paper presented a comprehensive literature review of the fundamentals of the continuum damage approach employed in the characterization of the fatigue resistance of asphalt mixtures. It also presented results of a laboratory study carried out to evaluate to effects of the addition of different proportions of reclaimed asphalt pavements (RAP) and two asphalt binders of distinct performance grades on the fatigue response of fine aggregate matrix (FAM) mixtures. In this study, the simplified VECD (S-VECD) theory was applied to analyze the resulting data from the damage tests, along with a detailed description of the calculations as an illustrative example of how to handle the equations of the S-VECD theory for this sort of analysis.

- The findings of the experimental study with RAP and binders PG 64-22 and PG 58-16 indicated that the FAMs containing 40% of RAP (2 and 4) presented higher $|G^*|_{LVE}$ values and higher damage evolution ratios as compared to the FAMs containing 20% of RAP (1 and 3). Out of the FAMs prepared with 20% of RAP (FAM1 and FAM3), the highest $|G^*|_{LVE}$ was observed for the FAM containing binder PG 64-22 (FAM1), and the damage evolution ratios were the same for both FAMs, which was an expected result, once the presence of the softest binder (PG 58-16) was supposed to reduce the stiffness of the FAMs (3 and 4).
- Regarding the prediction of the fatigue lives of the materials evaluated in the experimental study, the addition of RAP increased the parameter A of the fatigue model (related to the initial stiffness of the material and how the stiffness changed with

the evolution of the damage) and the parameter B (related to the damage evolution rate)—the resulting fatigue lives of the FAMs prepared with 20% RAP were longer than the ones obtained for the FAMs prepared with 40% of RAP. The fatigue performance was directly related to the specimen stiffness: the higher the stiffness, the higher its susceptibility to damage and the lower the relaxation rates (which resulted in higher damage accumulation rates). The best solution to adjust the binder content of FAMs produced with 20% and 40% of RAP was the use of the binder PG 58-16. The FAM tests combined with the S-VECD theory as a tool to analyze the results was a practical approach, and is widely used to evaluate all sorts of variables of an asphalt mixture. However, some variables, such as low temperatures and/or high percentages of RAP, turn the mixtures into overly stiff materials, and the tests can be unpractical due to the limits of the rheometer torque. Equipment with a higher torque capability could accelerate the test duration.

- The improvement of computational simulations of the test protocols is an important subject for future works, and could contribute to a better understanding of the mechanisms and variables involved in the fatigue process, and could also help overcome the rheometer limitations.

- Comparisons between fractures mechanics and continuum mechanics results could also be an interesting topic to improve the VECD model in order to account for the different types of damage: adhesive or cohesive.

- Regarding materials science and development of advanced/new materials, the FAM approach combined with the S-VECD approach offered several new possibilities in terms of material performance evaluation and material development. Some examples can be mentioned concerning the fatigue performance: (i) the evaluation of the impact of higher RAP contents added to new AC mixtures; (ii) the evaluation of the impact of recycling agents at different contents, including petroleum-based materials, vegetable-based oils, and recycled oils; (iii) the assessment of the aging impact on fatigue; (iv) the assessment of moisture damage on fatigue resistance; (v) the assessment of new asphalt modifiers, including hybrid modification using virgin and recycled materials; and (vi) the evaluation of the effect of distinct aggregate types and aggregate gradations, among others. Several doubts related to these subjects can be countered by carrying out tests at the FAM scale and using the S-VECD approach. However, one must keep in mind that such a development also depends on a larger number of experiments on the correlation between the fatigue performance at the two scales (FAM and full asphalt mixtures). Such experiments are essential for the development and popularization of these very promising techniques.

**Author Contributions:** Conceptualization, A.K. and A.N.; methodology, A.N.; software, A.K. and A.N.; validation, A.K., A.N. and A.F.; formal analysis, A.K. and A.N.; investigation, A.K.; resources, A.K. and A.F.; data curation, A.K. and A.N.; writing—original draft preparation, A.K. and A.N.; writing—review and editi A.K. and A.F.; visualization, A.K.; supervision, A.F.; project administration, A.F.; funding acquisition, A.F. All authors have read and agreed to the published version of the manuscript."

**Funding:** This research was funded by Coordenação de Aperfeiçoamento de Pessoal de Nível Superior (CAPES) under grant number 1770965. This research was also funded by the São Paulo State Research Funding Foundation (FAPESP) under grant number 12#2015/24082-1.

**Institutional Review Board Statement:** Not applicable.

**Informed Consent Statement:** Not applicable.

**Data Availability Statement:** Not applicable.

**Conflicts of Interest:** The authors declare no conflict of interest.

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
