# Peer review of "Application of the Viscoelastic Continuum Damage Theory to Study the Fatigue Performance of Asphalt Mixtures—A Literature Review"

_sustainability, doi:10.3390/su14094973_

Round 1

Reviewer 1 Report

Reviewers' comments:

Reviewer: Sustainability 2022,14 x for peer review

In the study, the authors included an extensive literature review that examined the application of the VECD theory to study the fatigue performance of asphalt mixture. It was submitted the fundamentals of the continuum damage approach performed in the characterization of the fatigue resistance of FAMs. Within the study, not only literature data was used but also laboratory experiments were carried out in the manuscript. It was used the fundamental of the VECD model and S-VECD applied to damage characterization of asphalt mixtures. In the experimental program, two different binders (PG58-16 and PG 64-22) and two RAP proportions (20%, 40%) were performed to evaluate the fatigue performance of FAMs.

According to the experimental study, fatigue performance was affected by rigidity and the binder content of FAMs produced with 20% and 40% of RAP were come out of the binder PG58-18, to get the best solution. Furthermore, fatigue lives of the FAMS which are prepared with 20% RAP have appeared longer than 40% RAP of FAMs

Concerning the aforementioned manuscript, the following query and advice should be considered.

Queries and advice

1-     Have many samples have been using for experimental study?

2-     Where did bitumen and aggregate accommodate for the test process?

3-     Where did the study conducted for materials?

4-     The experimental method used in the study should be explained in the purpose section.

5-     The path followed in the study should be briefly summarized and a flow chart should be added to the manuscript.

6-     The device types and software program version used in the studies with brief technical information should be given in the manuscript.

7-    The authors must be pointed out the physical properties of neat and modified asphalt on the table.

8-    Aggregates should be submitted with physical and mechanical properties.

9-    Current citations focusing on the aim of the study should also be included in the article.

10-  In the manuscript, the missing parts of the semantic integrity should be corrected (lines 125-130).

11- Some abbreviations should be explained due to not explaining some parts of the manuscript.

12- What is the Witco AR-400? If it is a brand name, it should not be included in the manuscript.

13- Abbreviations in the formula should be explained (for example equation 12).

14- Some titles are too long. The contents of these titles should be shortened. (3.1; 3.6 etc.)

15- A brief explanation should be made about the Maxwell model, Prony series, Scharepy.

16- Some references are over-emphasized. These parts should be simplified.

17- There are too many references to glass beads.

18- There are references to numerical analysis in the manuscript, but the authors do not have any studies.

19- There is no need for references to the warm aggregate.

20- Table numbering is incorrect. The references given regarding the table number should be corrected.

21- The formulas cited in the table and the ones given in the article are not compatible with each other.

22- Data sensitivity should be increased to better understand the difference in Figure 2.

23- Table 4 is mentioned in the study. But it is not included.

24- Upper and lower threshold values should be given in Figure 3.

25- The ranges should be revised to better understand the curve differences in Figure 4.

26- Some color scales in the figures are very close to each other. Contrast colors should be chosen to understand the difference.

27- Function equations and correlation coefficients should be added to the graphs as tables.

28- Some references are too long. Detailed tests are explained. Focus on results and make abbreviations.

29- An experimental work flowchart should be shown.

30 - Repeated sentences should be avoided.

31- How does this study benefit the literature when compared to citations?

As a result of the above evaluations, numerous citations are included in the study. The content of some of the citations is too long. There are errors in the formula contents of the equations. Laboratory studies are also included, but the study is too complicated, the long definition and the content of some references are kept too long, and too many citations show similarities in the literature intensely. it appears that the manuscript is understandable in terms of its structure, fiction if the above recommendation must be considered. A study of literature can be thought of as a book chapter rather than a manuscript. The main thing in the manuscript, the plagiarism similarities of the manuscript are too many. Because of that, the review opinion the manuscript is not acceptable for this journal.

Reviewer 2 Report

Dear authors,

Thanks for submitting your paper to the sustainability journal. The review topic is an interesting topic, however, I have  few comments that should be used to improve the final version of the paper;

1- Why viscoelastic continuum damage (VECD) model? why this model? give reason to clarify the selection, Alos I suggest to illustrate a table and show a comparison between this model and other models (properties ), and what is the benefits of selecting this model. 

2- must showr3search gap in the separated table. 

3- must show a flow chart of the review strategies of this paper, and contest to make it a more scientific presentation. 

4- add references to each table/figure used in this paper.

5- please make the conclusion as point by point to address the paper's objectives. and add recommendations for future work on this topic. 

all the best.

Reviewer 3 Report

This paper presents application of the viscoelastic continuum damage theory to study the fatigue performance of asphalt mixtures. As a literature review, the whole manuscript especially the third part should be reorganized to be more logical, making it better to give a whole picture of this research field.

  1. What’s the problems in application of the viscoelastic continuum damage theory to study

the fatigue performance of asphalt mixtures?

  1. Try to compare with the Burgers Model in “ZHANG Xiao-de, ZHANG Jun. Fatigue Damage Evolution Characteristics and Model of Asphalt Mixture Based on Energy Method ï¼»Dï¼½. Shenyang: Northeastern University, 2017.”
  2. On Line 50 and 51, “The data obtained in tests performed with both FAMs and full asphalt mixtures can be analyzed by means of the continuum mechanics theory.

The reason why asphalt mixtures can be analyzed by means of the continuum mechanics theory is not mentioned in this paper. Sentences like “As asphalt mixture is a typical viscoelastic or viscoelastic-plastic material…… ” should be added.

  1. Some descriptions are uncertain and confusing, for example,

on Line 43, “It is hypothesized that studies……”;

on Line 70, “In an attempt to handle the material variability”.

  1. The English language of the manuscript should be carefully revised. For example,

on Line 312, “at different load and environmental conditions ”, load should be replaced by loads;

on Line 453, “Based on previous studies developed in order to characterize the damage behavior”;

on Line 457, “which consists in two steps ”, in should be replaced by of.

on Line 673, “indicating that that the WMA mixtures”.

on Line 855-857, “In summary, the increase in the RAP content from 20 to 40% increased the fatigue life at low strains only if the binder content is corrected with a binder PG 64-22 and reduced it at high strains.”, the tense is wrong.

  1. On Line 525, “1. Fine aggregate matrix approach”, but where is 3.2?
  2. On Line 542-547, the addition is verbose as it has been explained in the introduction part on Line 43-49.

On Line 923-932, the description is verbose in the conclusions part.

  1. On Line 724, “in an especially designed mold”, the mold should be explained with clear experimental program.

Reviewer 4 Report

These are reviewers comments for the manuscript entitled: Application of the viscoelastic continuum damage theory to study the fatigue performance of asphalt mixtures – A literature review, submitted for publication in MDPI Sustainability. 

The work presented in the manuscript is very interesting however I do not think that it is out of the scope of the MDPI Sustainability journal. I believe that there are more suitable journals for publication of this paper, such as MDPI Materials or one of the special issues journals. I am recommending for the paper to be submitted to a more suitable journal.  

My further comments on the paper are:

  1. Authors call the paper review, if so what is the reason for the review, if authors want to discuss the fatigue numerical models then wider review should be taken. I believe that this should be focused on the damage model and summerised. 
  2. paper is far too long sections 2-4 are too long and read as a extractions form academic thesis or a book. They should be summeriesed and merged with the Introduction.
  3. Introduction, pg 1 - 2, authors make a lot of statements and arguments without supporting them with background literature - where these statements come from. It reads more like ones opinion rather than scientific review. 
  4. Sections 2,3 and 4 should be summerised and authors should focus on the S-VECD theory applied in their study. 
  5. Section 7 - Remove it
  6. Acronym section - is this required by the journal if so OK if not remove it and include acronym definitions in the main text. 

Round 2

Reviewer 1 Report

I check out the manuscript again after the first review.
The authors have given some query's answer and it is necessary to consider the following things.
1- Aggregate and bitumen standards should be added to the manuscript.
2- Some word was used by the authors like baderientes? What does it mean? ıt must be removed from the manuscript.

As a result, a literature study was carried out and some corrections were made by the authors.   however, the similarities abound as a general template due to the literature.  Adjustments should be made taking into account the corrections I have mentioned.

Author Response

Dear reviewer,

Thank you very much for all your comments. Please see the attachment.

Reviewer 2 Report

Accepted.

Author Response

Dear Reviewer,

Thank you for your comments.

Best regards,

Ms. Andrise Klug.

Reviewer 3 Report

Dear authors,

Thanks for answering and solving most of the questions raised last time. My further comments on the revised paper are:

  1. According to your choosing to use the Maxwell Model, please show a comparison between this model and other models (like Burgers Model). 
  2. The following are different questions from me and another reviewer, but your two replys are the same!

3.The English language of the manuscript should be carefully revised again .For example,

on Line 324“at different load and environmental conditions ”, load should be replaced by loads.

Author Response

(The authors gave the same response as above.)

Reviewer 4 Report

Dear authors thank you for taking comments on board. Though I still believe that paper is more suitable to journal such as: MDPI Materials or  MDPI Applied Sciences, Special Issue: Advances in Asphalt pavement Technologies and Practices. Reason for this comment is that work focuses on material performance simulation and doesn't really addresses any sustainability issues such as use of greet technologies and approaches in asphalt mix design. Which is out of the scope of the Sustainability paper. In my opinion applying the numerical model to predict asphalt (containing RAP) material performance does not count as sustainability project. I suggest transfer for the paper to more suitable journal.

My further comments and suggestion to authors are: 

  1. Paper is still far too long Section two should be summarised and focused on discussion of VECD model.
  2. Authors still keep making statements in the introduction section without referencing to source material such as: Introduction ln 26 - 28 ,Authors state that predicting fatigue behaviour of asphalt mixtures has been the goal of many studies, please give examples - references.
  3. ln 28 - 31 Authors give an example of phenomenological and mechanistic models, though fail to reference source material -  please include reference to source material where there statement/conclusions originate from.
  4. l36 - 39 again authors need to add references to these statements.
  5. ln 53 - 59 discussion about VECD is missing references where these definitions and statements come from? This is review paper?
  6. Figure 1 pg 2. figure heading needs to be corrected.

Author Response

Dear Reviewer,

Thank you very much for all your comments. Please see the attachment.

Best regards,

Ms. Andrise Klug
